# Greedy Multi-Path Block Verification for Faster Decoding in Speculative Sampling

## Abstract

The goal of $L$-step speculative decoding is to accelerate autoregressive decoding of a target model by using a cheaper draft model to generate a candidate path of $L$ tokens. Based on a verification algorithm involving target and draft model probabilities, a prefix of the candidate sequence is accepted, and an additional correction token is sampled from a residual distribution to ensure that the final output adheres to the target distribution. While standard speculative decoding uses a verification algorithm which is independent at each token on the path, a recent extension called block verification uses a joint condition involving all sampled on-path probabilities. Block verification (BV) was shown to be optimal over all verification algorithms which use only on-path probabilities, improving on standard speculative decoding. In this work, we first show that block verification is optimal even over verification algorithms that use off-path probabilities, by constructing an information-agnostic linear program (LP). Further, we can extend our LP to the setting where the draft model samples multiple candidate paths, and use it to construct a natural class of multi-path block verification generalizations. While computing the optimal algorithm in this class is not tractable, by considering a stricter class of greedy algorithms, we can formulate an efficient method called greedy multi-path block verification (GBV). Empirically, GBV can improve block efficiency by over 30% and reduce decoding walltimes by over 15% relative to BV.

## 1 Introduction

Large language models (LLMs) achieve strong results across code, language, and reasoning (Zhu et al., 2024; Kasneci et al., 2023; Thirunavukarasu et al., 2023). Most LLM families, such as Qwen (Bai et al., 2023), GPT (Radford et al., 2018; 2019; Brown et al., 2020; OpenAI, 2023), and Llama (Touvron et al., 2023b;a) employ autoregressive decoding. When inference is performed with these transformer-based architectures on GPUs, end-to-end latency is dominated by memory bandwidth rather than compute (Fu et al., 2024).

Speculative sampling (Chen et al., 2023; Leviathan et al., 2023) aims to reduce such punitive costs when sampling from a large *target model*. This procedure autoregressively decodes a candidate sequence from a cheaper *draft model*, performs a forward pass over the candidate sequence with the target model to obtain target distribution values, and then probabilistically alters the sequence through a *verification algorithm* to ensure that the resulting output matches the target model distribution. Because inference is bandwidth-bound, the target forward pass over the candidate sequence presents negligible overhead over a standard forward pass. Thus, speculative sampling can decode many tokens in a single target model call and speed up inference without affecting downstream performance.

In **standard speculative sampling**, the draft model proposes a length $L$ draft block, and the verification algorithm independently accepts or rejects each token based on target and draft model distributions values along the draft bock. The longest prefix with no rejections is chosen, and an additional token is sampled from a residual distribution, ensuring that at least one token is always generated. While this procedure significantly improves decoding efficiency, it suffers when there are low acceptance rates at early tokens. If the first token is almost always rejected due to a poor draft suggestion, then even if subsequent tokens are always accepted, there is no speedup.

To overcome this early-token bottleneck, recent work on **block verification (BV)** (Sun et al., 2024b) replaces independent token-wise verification with independent prefix-wise verification, and selects

the longest accepted prefix. An additional token is still sampled, but the residual distribution is altered to ensure the output still matches the target distribution. This ensures that more tokens can be accepted when earlier tokens are likely to be rejected. Block verification is empirically 5-8% faster than standard speculative sampling. In fact, it is provably optimal over all verification algorithms that only use target and draft probability distributions along the drafted candidate sequence, including standard speculative sampling.

In this paper, we frame the optimal verification algorithm as a solution to a linear program (LP), and extend the LP to the setting of *multiple* i.i.d. generated draft sequences. Our contributions are:

- **Single-Path Information-Agnostic LP.** We develop an LP that encodes feasible speedups from verification algorithms that must still return a candidate sequence prefix and an additional token, but are now given access to the *full joint* target and draft distributions, not just on-path values. We show that block verification is still optimal in this setting. That is, the prefix output requirement is the bottleneck to improving optimal decoding efficiency, rather than off-path information access.

- **Multi-Path Information-Agnostic LP.** We extend the single-path LP to the setting where the draft model i.i.d. samples $K > 1$ candidate sequences, considering verification algorithms which return a prefix of one of the paths and an additional token. Unlike the $K = 1$ setting, we find that knowledge of off-path target and draft distributions can improve decoding efficiency. We show that the optimal verification algorithm in this setting is to probabilistically select one of the $K$ paths, and then run block verification on that single path with a skewed draft distribution.

- **Greedy Approximation Schemes.** The optimal solution to the multi-path LP involves a complex nonlinear optimization problem over the joint target and draft distributions, and is infeasible to solve directly. Thus, we explore a class of greedy approximations: algorithms which globally rank all possible candidate sequences, select the highest-ranking of the $K$ paths, and run block verification on that path. We show that these can be viewed as outputs of a greedy polymatroid algorithm in the multi-path LP.

- **Greedy Multi-Path Block Verification (GBV).** Many greedy approximation schemes require the full joint target and draft distributions. However, when the global ranking is tree-based, only on-path values are required. We devise a simple tree-based rule that results in wall-clock speedups of over 15% relative to BV, and provide theoretical justification for this rule.

## 2 BACKGROUND

We first review standard speculative sampling and its extensions. We start with the case where only one draft block is generated, and then cover multi-path extensions.

### 2.1 SINGLE-PATH

Denote the target model by $M_p$, and the draft model by $M_q$, which are assumed to share a common vocabulary $\mathcal{V}$. We use the sequence notation $a_{1:k} = (a_1, \ldots, a_k)$ for tokens $a_1, \ldots, a_k \in \mathcal{V}$ and the inclusive slicing notation $a_{i:j} = (a_i, \ldots, a_j)$, with the empty sequence convention $a_{i,j} = \emptyset$ for $j < i$. Given a context $\boldsymbol{c}$, the target and draft models induce next-token distribution $p(\cdot|\boldsymbol{c})$ and $q(\cdot|\boldsymbol{c})$, respectively. In autoregressive sampling, these further induce next-$k$-token distributions through conditional factorization, which we denote:

$$q_k(a_{1:k}|\boldsymbol{c}) = \prod_{i=1}^{k} q(a_i|\boldsymbol{c}, a_{1:i-1}), \quad p_k(a_{1:k}|\boldsymbol{c}) = \prod_{i=1}^{k} p(a_i|\boldsymbol{c}, a_{1:i-1}). \quad (1)$$

In $L$-step speculative sampling, we autoregressively sample a draft block of $L$ tokens from the draft distribution $q_L$ and call the target model $M_p$ on this block *once*, to obtain target and draft next-token probabilities along the block. Through a randomized rule called the **verification algorithm**, we accept the first $\tau \in \{0, \ldots, L\}$ of these $L$ tokens and sample an additional correction token, such that this output matches the true target distribution. The **block efficiency** $\mathbb{E}[\tau + 1]$ is the average number of decoded tokens per $M_p$ call; in the special case where the target and draft distributions are

identical, it achieves its maximum value of $L + 1$. When a cheap draft model is used and $L$ is not too large, block efficiency is an accurate indicator of walltime speedup.

**Speculative sampling.** The original schemes of Chen et al. (2023); Leviathan et al. (2023) first generate a draft block $a_{1:L} \sim q_L(\cdot|\boldsymbol{c})$. They *independently* accept each token $a_i$ with probability $\min(1, p(a_i|a_{1:i-1})/q(a_i|a_{1:i-1}))$, and $\tau$ is the maximum value such that $a_{1:\tau}$ consists of only accepted tokens. Essentially, they choose the longest prefix of acceptances. Finally, they sample an additional *correction token* $y \sim p_{\text{res}}^{\text{token}}(\cdot|\boldsymbol{c}, a_{1:i})$ from a residual distribution if $\tau < L$:

$$p_{\text{res}}^{\text{token}}(\cdot|\boldsymbol{c}, a_{1:i}) \propto \max\{p(\cdot|\boldsymbol{c}, a_{1:i}) - q(\cdot|\boldsymbol{c}, a_{1:i}), 0\}. \tag{2}$$

If $\tau = L$, they just sample $y \sim p(\cdot|\boldsymbol{c}, a_{1:L})$ directly from the target. This ensures that the final output $(a_{1:\tau}, y)$ matches the target distribution.

**Draft extensions.** Some recent works have improved upon block efficiency in speculative sampling by altering the drafting phase, through retrieval or cascading (He et al., 2023; Chen et al., 2024), hierarchical drafting (Sun et al., 2024a), distillation (Zhou et al., 2023; Liu et al., 2023), layer skipping (Zhang et al., 2023; Elhoushi et al., 2024), or multi-token heads (Gloeckle et al., 2024; Samragh et al., 2025)). In this paper, we instead focus on methods which improve block efficiency by altering the verification algorithm, and these can be integrated with any of the above works for further gains.

**Tree verification.** Monte Carlo tree verification from Hu & Huang (2024) provably improves the block efficiency of standard speculative sampling. While the underlying tree structure might suggest this is a multi-path method, we emphasize that it is a single-path algorithm, as the end of Section 5 in Hu & Huang (2024) specifies a linear chain of draft tokens. This means it is provably worse than block verification: see the third paragraph of Section 7 in Sun et al. (2024b).

**Block verification (BV).** To improve block efficiency in the verification stage, Sun et al. (2024b) relax the token-independence assumption for acceptance in standard speculative sampling. First, they sample the draft block $a_{1:L} \sim q_L(\cdot|\boldsymbol{c})$, and recursively define on-path weights $w$:

$$w(\emptyset|\boldsymbol{c}) = 1, \quad w(a_{1:i}|\boldsymbol{c}) = \min\left\{1, \frac{w(a_{1:i-1}|\boldsymbol{c})p(a_i|a_{1:i-1})}{q(a_i|a_{1:i-1})}\right\}. \tag{3}$$

Now, they independently accept each *prefix* $a_{1:i}$ with probability

$$h^{\text{block}}(a_{1:i}|\boldsymbol{c}) = \frac{\sum_{x \in \mathcal{V}} \max\{p(x|\boldsymbol{c}, a_{1:i}) - q(x|\boldsymbol{c}, a_{1:i}), 0\}}{1 - w(a_{1:i}|\boldsymbol{c}) + \sum_{x \in \mathcal{V}} \max\{p(x|\boldsymbol{c}, a_{1:i}) - q(x|\boldsymbol{c}, a_{1:i}), 0\}} \tag{4}$$

for $i < L$, and $a_{1:L}$ with probability $w(a_{1:L}|\boldsymbol{c})$. They select the longest accepted prefix, of length $\tau$, and sample a correction token $y \sim p_{\text{res}}^{\text{block}}(\cdot|\boldsymbol{c}, a_{1:i})$, a weighted version of $p_{\text{res}}^{\text{token}}$, if $\tau < L$:

$$p_{\text{res}}^{\text{block}}(\cdot|\boldsymbol{c}, a_{1:i}) \propto \max\{w(a_{1:i}|\boldsymbol{c})p(\cdot|\boldsymbol{c}, a_{1:i}) - q(\cdot|\boldsymbol{c}, a_{1:i}), 0\}. \tag{5}$$

If $\tau = L$, they sample $y \sim p(\cdot|\boldsymbol{c}, a_{1:L})$. Like in speculative sampling, the output $(a_{1:\tau}, y)$ follows the target distribution. Sun et al. (2024b) prove that block verification achieves the highest block efficiency among any verification algorithm that only takes in on-path distribution values.

## 2.2 Multi-Path

There are also recent lines of work that extend $L$-step speculative sampling to the *multi-path* setting (Sun et al., 2023; Spector & Re, 2023; Cai et al., 2024; Li et al., 2024). In this setting, $K > 1$ length-$L$ draft blocks are sampled from $q_L$. The target model $M_p$ is again called *once* on all draft blocks in parallel, to obtain target and draft next-token probabilities along all $K$ paths. Then, using a **multi-path verification algorithm**, a prefix of one of the blocks is accepted, and an additional correction token from a residual distribution is sampled in order to match the target distribution. Block efficiency is defined in the same way as previously. For moderate $K$ and $L$, leveraging GPU parallelization, there is generally little overhead in performing the *batched* forward pass across $K$ sequences, relative to a forward pass on one sequence (Agrawal et al., 2024; Dao et al., 2022).

We use the concept of a **draft tree** to enumerate all distinct prefixes among the $K$ sampled paths. We will compare our methods against two categories of multi-path verification algorithms which select a node from the draft tree, i.e. a prefix of one of the blocks, in different ways.

**OTLP-based algorithms.** These perform a top-down traversal of the draft tree. At each step, they compute a feasible solution of an optimal transport linear program (OTLP) to determine the next token and progress to a child node, terminating when they land off the draft tree. SpecTr (Sun et al., 2023), NSS (Miao et al., 2024), and SpecInfer (Miao et al., 2024) are key examples, and recent theoretical OTLP solvers (Khisti et al., 2025; Hu et al., 2025) further improve block efficiency.

**Traversal verification.** To the best of our knowledge, traversal verification in Weng et al. (2025) is the only multi-path algorithm that traverses the draft tree from the bottom-up. Like our algorithm in Section 5, this reduces to BV in the single-path $K = 1$ case. In our experiments, traversal verification and our methods outperform OTLP-based methods, but are each effective in different $(p, q)$ regimes.

## 3 SINGLE-PATH INFORMATION-AGNOSTIC LP

We first formally define the class of single-path verification algorithms. We denote $\mathcal{V}^{\leq k} = \mathcal{V}^0 \cup \mathcal{V}^1 \cup \ldots \cup \mathcal{V}^k$, where $\mathcal{V}^k$ is the set of length-$k$ sequences of tokens in $\mathcal{V}$, and thus $\mathcal{V}^{\leq k}$ is the set of sequences of length $\leq k$. We use $L$ to denote the draft block length. For the remainder of the paper, we use notation from Section 2, omitting the context $c$ and subscripts on $p, q$ when they are clear.

**Definition 3.1** (Single-path draft verification algorithm). A **single-path verification algorithm** $\Phi$ takes in a sampled draft block $X_{1:L} \sim q_L(\cdot)$, and the full $L$-step target and draft distributions $p_L$ and $q_L$, and returns a nonempty sequence in $\mathcal{V}^{\leq L+1}$. It is **valid** under:

- **Prefix-matching:** the algorithm returns a (possibly empty) prefix $X_{1:\tau}$ of the draft block followed by an additional correction token $Y$, for some $\tau \in \{0, \ldots, L\}$.

- **Target-matching:** for any context $c$, any draft and target models $M_p$ and $M_q$, and any block length $L$, when we generate $L - \tau$ additional tokens $Z \sim p_{L-\tau}(\cdot|X_{1:\tau}, Y)$, we have $(X_{1:\tau}, Y, Z) \sim_p p_{L+1}(\cdot)$, where $\sim_p$ denotes equality of distributions.

If we also require that $\Phi$ can only take in the *on-path* target and draft distributions $p(\cdot|X_{1:i}), q(\cdot|X_{1:i})$ for $i \in \{0, \ldots, L\}$, then it is called **information-restricted**.

Target-matching means that sampling from $M_p$ autoregressively after verification, until a total of $L+1$ tokens are generated, gives the same result as just sampling $L + 1$ tokens from $M_p$ autoregressively without verification. This is equivalent to guaranteeing that running the verification algorithm and appending its output to the context iteratively maintains the target distribution, even as $\tau$ varies: see Lemma 2 in Appendix B of Sun et al. (2024b) for a formal proof.

Prefix-matching forces the verification output to only deviate from the draft block at its last token. This is a strict modeling requirement, not a technical convenience. To bypass prefix-matching while maintaining the target distribution, one must access the next-token target distribution $p(\cdot|c)$ for contexts $c$ outside the draft tree. This is not possible without performing another $M_p$ forward pass, which would defeat the goal of speculative decoding: generating multiple tokens in one target call.

Importantly, our definition is *less strict* than in Sun et al. (2024b), because they only consider the class of valid single-path *information-restricted* verification algorithms, and prove that block verification is optimal in this class. Surprisingly, we find that block verification is optimal in the class of *all* valid single-path verification algorithms. To prove this, we first define node budgets, which represent how much mass a verification algorithm has allocated along a path relative to the target distribution.

**Definition 3.2.** For any single-path verification algorithm $\Phi$, we define **node budgets**

$$D_\Phi(a_{1:i}) = 1 - \sum_{j=1}^{i} \frac{\mathbb{P}(X_{1:\tau} = a_{1:j-1}, Y = a_j)}{p(a_{1:j})} \quad \forall a_{1:i} \in \mathcal{V}^{\leq L+1}. \tag{6}$$

This can be represented by a function $D_\Phi : \mathcal{V}^{\leq L+1} \to \mathbb{R}$.

Using node budget variables, we define the **single-path information-agnostic LP** in Theorem 3.3, which describes what values of $D_\Phi$ a valid single-path verification algorithm $\Phi$ can induce. Here, the chain of $D_\Phi$ inequalities along paths encode the target-matching constraint, and the pointwise upper bounds on $D_\Phi p$ encode prefix-matching. We defer the proof to Appendix B, as the special case $K = 1$ of the multi-path LP in Theorem 4.3, which we prove in Appendix A.

**Theorem 3.3.** *A single-path verification algorithm $\Phi$ is valid if and only if the node budgets $D_\Phi : \mathcal{V}^{\leq L+1} \to \mathbb{R}$ are* feasible *in the following LP:*

$$\max \sum_{a_{1:i} \in \mathcal{V}^{\leq L}} D_\Phi(a_{1:i})p(a_{1:i}) \tag{7}$$

$$s.t. \quad 1 = D_\Phi(a_{1:0}) \geq D_\Phi(a_{1:1}) \geq \ldots \geq D_\Phi(a_{1:L}) \geq D_\Phi(a_{1:L+1}) = 0 \quad \forall a_{1:L+1} \in \mathcal{V}^{L+1}, \tag{8}$$

$$D_\Phi(a_{1:i})p(a_{1:i}) \leq q(a_{1:i}) \quad \forall a_{1:i} \in \mathcal{V}^{\leq L}. \tag{9}$$

*Furthermore, the objective value is precisely the block efficiency $\mathbb{E}[\tau + 1]$ for $\Phi$.*

Because the feasibility conditions in the single-path LP only involve pointwise and pathwise inequalities, it is not hard to compute the optimal objective value and node budgets for a valid single-path verification algorithm, by using greedy allocation along paths. These node budgets match those derived from block verification, and thus block verification is the optimal valid single-path algorithm.

**Theorem 3.4.** *Block verification, which is a valid single-path verification algorithm $\Phi_{BV}$, achieves the highest block efficiency of any valid single-path verification algorithm.*

See Appendix C for a proof of Theorem 3.4. These results show that prefix-matching is the key barrier to improving acceptance: even with access to the full joint target and draft distributions, prefix-matching (pointwise upper bounds on $D_\Phi p$ terms) prevent us from getting better block efficiency (the sum of $D_\Phi p$ terms) than block verification. This motivates our exploration of verification algorithms which draft *multiple* candidate paths in the next section. In this setting, the valid prefix output space grows, thereby loosening the prefix-matching requirement and improving block efficiency.

## 4 MULTI-PATH INFORMATION AGNOSTIC LP

In this section, we show that off-path distribution information can improve block efficiency when $K > 1$ draft paths are generated, even with prefix-matching and target-matching requirements. We first define the class of multi-path verification algorithms, and extend Theorem 3.3 to these algorithms.

**Definition 4.1** ($K$-path draft verification algorithm). A $K$-path verification algorithm $\Phi$ takes in $K$ i.i.d. sampled draft blocks $X_{1:L}^{(1)}, \ldots, X_{1:L}^{(K)} \sim q_L(\cdot|\boldsymbol{c})$, and the full $L$-step distributions $p_L$ and $q_L$, and returns a nonempty sequence in $\mathcal{V}^{\leq L+1}$. It is **valid** under:

- **Prefix-matching:** the algorithm returns a (possibly empty) prefix $X_{1:\tau}$ of *some* draft block followed by an additional correction token $Y$, for some $\tau \in \{0, \ldots, L\}$.

- **Target-matching:** this is the same as in Definition 3.1.

If $\Phi$ can only take in the *on-path* target and draft distributions $p(\cdot|X_{1:i}^{(k)}), q(\cdot|X_{1:i}^{(k)})$ for $k \in [K]$ and $i \in \{0, \ldots, L\}, k \in [K]$, then it is called **information-restricted**.

To the best of our knowledge, all existing valid multi-path verification algorithms (see Section 2.2) are information-restricted. We are the first to consider theoretical efficiency limits in the absence of information-restriction, and explicitly encode prefix-matching and target-matching into an LP.

To extend the single-path LP from Theorem 3.3 to this setting, we first define node budgets $D_\Phi$ in the same way as Definition 3.2. We also require the notion of an *antichain*, which is a set of variable-length paths in $\mathcal{V}^{\leq L}$ where no path is a prefix of another. Antichains are useful because the mass of an autoregressive distribution over an antichain can be computed by summing its probabilities at all antichain elements, due to the disjointness of autoregressively sampling antichain elements.

**Definition 4.2.** An **antichain** of $\mathcal{V}^{\leq L}$ is a subset of $\mathcal{V}^{\leq L}$ where no sequence in the antichain is a prefix of another. We denote the set of all antichains in $\mathcal{V}^{\leq L}$ by $\mathcal{A}(\mathcal{V}^{\leq L})$.

Now, to form the **multi-path information-agnostic LP** in Theorem 4.3, we replace the pointwise upper bounds on $D_\Phi p$ by subset-sum upper bounds over antichains. As mentioned at the end of Section 3, this corresponds to altering the prefix output space. The proof uses Hall-type feasibility constraints, and is fairly involved. Due to space constraints, we defer the proof to Appendix A.

**Theorem 4.3.** *A $K$-path verification algorithm $\Phi$ is valid if and only if the node budgets $D_\Phi$ : $\mathcal{V}^{\leq L+1} \to \mathbb{R}$ are feasible in the following LP:*

$$\max \sum_{a_{1:i} \in \mathcal{V}^{\leq L}} D_\Phi(a_{1:i})p(a_{1:i}) \tag{10}$$

$$s.t. \quad 1 = D_\Phi(a_{1:0}) \geq D_\Phi(a_{1:1}) \geq \ldots \geq D_\Phi(a_{1:L}) \geq D_\Phi(a_{1:L+1}) = 0 \quad \forall a_{1:L+1} \in \mathcal{V}^{L+1}, \tag{11}$$

$$\sum_{a_{1:i} \in T} D_\Phi(a_{1:i})p(a_{1:i}) \leq 1 - \left(1 - \sum_{a_{1:i} \in T} q(a_{1:i})\right)^K \quad \forall T \in \mathcal{A}(\mathcal{V}^{\leq L}). \tag{12}$$

*Furthermore, the objective value is precisely the block efficiency $\mathbb{E}[\tau + 1]$ for $\Phi$.*

This reduces to the single-path LP in the case $K = 1$. However, for $K > 1$, there is a important distinction: the upper bound on sums of $D_\Phi p$ now represent a *submodular* function of $T$, unlike the *additive (modular)* pointwise upper bounds on $D_\Phi p$ in Theorem 3.3. We now explain how this submodularity naturally arises from a **path selection rule** in valid $K$-path verification algorithms. In Lemma 4.4 (proof in Appendix D), we first prove a canonical decomposition of such algorithms into randomized selection of one of the drafted $K$ paths, followed by valid single-path verification.

**Lemma 4.4.** *A valid $K$-path verification algorithm has the following equivalent definition. First, given $K$ paths sampled i.i.d. from $q$, randomly select one through a path selection rule $\Gamma$, which can depend on off-path probability values. Say this path follows the distribution $q^\Gamma$ over $\mathcal{V}^L$. Then, run a valid single-path verification algorithm on this path with target values $p$ and draft values $q^\Gamma$.*

We call $q^\Gamma$ the **skewed draft distribution** induced by $\Gamma$. Because this is a distribution over $\mathcal{V}^L$, just as for $p_L, q_L$, we can naturally extend it to an induced distribution over each $\mathcal{V}^i$ for $0 \leq i \leq L$:

$$q^\Gamma(a_{1:i}) = \sum_{a_{i+1:L} \in \mathcal{V}^{L-i}} q^\Gamma(a_{1:L}) \quad \forall a_{1:i} \in \mathcal{V}^{\leq L}. \tag{13}$$

This further induces an autoregressive distribution in the same way as $p, q$:

$$q^\Gamma(a_{k+1:i}|a_{1:k}) = \frac{q^\Gamma(a_{1:i})}{q^\Gamma(a_{1:k})} \quad \forall a_{1:i} \in \mathcal{V}^{\leq L}. \tag{14}$$

Not all distributions can be realized as the skewed draft distribution of a path selection rule, given a fixed draft $q$ and path count $K$. In fact, as in Lemma 4.5, the realizable distributions are those which satisfy a submodular inequality like that in Theorem 4.3. For a proof, see Appendix E.

**Lemma 4.5.** *Fix $K$, a draft distribution $q$, and a distribution $r$. Then $r = q^\Gamma$ for some randomized path selection rule $\Gamma$ over $K$ drafts sampled i.i.d. from $q$ if and only if*

$$\sum_{a_{1:i} \in T} r(a_{1:L}) \leq 1 - \left(1 - \sum_{a_{1:i} \in T} q(a_{1:i})\right)^K \quad \forall T \in \mathcal{A}(\mathcal{V}^{\leq L}). \tag{15}$$

This shows that Lemma 4.4 corresponds to **linearizing** the submodular constraints in Theorem 4.3. Indeed, consider replacing the *submodular* upper bounds in the multi-path LP with the *additive (modular)* upper bounds $\sum_{a_{1:i} \in T} q^\Gamma(a_{1:i})$. By the upper bound in Lemma 4.5, any solution to this **linearized LP** will satisfy the original multi-path LP. In fact, this linearized LP is precisely the single-path LP[1] where the draft distribution is the skewed draft $q^\Gamma$ rather than the original $q$. This corresponds to running valid single-path verification with target $p$ and draft $q^\Gamma$, as in Lemma 4.4.

For a given $q^\Gamma$, using the result of Theorem 3.4, the optimal valid single-path verification algorithm to run here is precisely block verification. This leads to an explicit description of the optimal valid multi-path verification algorithm in Theorem 4.6. We prove this in Appendix F.

**Theorem 4.6.** *The optimal valid multi-path verification algorithm randomly chooses one of the $K$ i.i.d. blocks with $\Gamma$ and then runs single-path block verification on that path with target values $p$*

---

[1]See Appendix B for an explanation of how the sums reduce to pointwise inequalities like in Theorem 3.3.

*and draft values $q^\Gamma$, for some path selection rule $\Gamma$. Furthermore, the optimal choice of $\Gamma$ can be determined by solving the following optimization problem:*

$$\max_{a_{1:i} \in \mathcal{V}^{\leq L}} \sum \min_{0 \leq k \leq i} p(a_{k+1:i}|a_{1:k}) q^\Gamma(a_{1:k}) \tag{16}$$

$$s.t. \quad \sum_{a_{1:L} \in T} q^\Gamma(a_{1:L}) \leq 1 - \left(1 - \sum_{a_{1:L} \in T} q(a_{1:L})\right)^K \quad \forall T \subseteq \mathcal{V}^L. \tag{17}$$

This optimization problem is intractable to solve exactly, since it requires knowledge of the full joint and target distributions. Furthermore, even if the optimal $q^\Gamma$ can be computed, there is no guarantee that we can compute an efficient path selection rule $\Gamma$ which induces this skewed draft distribution. Therefore, we turn to approximation-based schemes. As Lemma 4.7 shows, we can obtain a near-optimal solution by making all ratios $q^\Gamma/p$ close to one. For a proof, see Appendix G.

**Lemma 4.7.** *For a fixed $q^\Gamma$, the objective value in Theorem 4.6 is lower bounded by*

$$(L+1) \cdot \min_{a_{1:i} \in \mathcal{V}^{\leq L}} \frac{q^\Gamma(a_{1:i})}{p(a_{1:i})} \tag{18}$$

This shows that when $q^\Gamma = p$ is feasible in Theorem 4.6, we can get an optimal block efficiency of $L+1$, i.e. we always accept a full length-$L$ path and an additional token. However, even determining a rule $\Gamma$ that achieves $q^\Gamma$ near $p$ is difficult. Thus, in the next section, we restrict ourselves to a more tractable class of *greedy* algorithms, where one can both explicitly compute the randomized rule $\Gamma$ and the resulting distribution values $q^\Gamma$. This explicit characterization is necessary to apply Lemma 4.4, because block verification requires exact draft probability values $q^\Gamma$ along the selected path, and it is impossible to exactly sample from $q^\Gamma$ without an explicit randomized rule $\Gamma$.

## 5 GREEDY MULTI-PATH BLOCK VERIFICATION

We now design an explicit path selection rule $\Gamma$ where the skewed draft distribution $q^\Gamma$ is heuristically close to $p$ and simple to compute, using only on-path probability values. This leads to our main algorithm: **greedy multi-path block verification (GBV)**. We first define greedy verification algorithms.

**Definition 5.1.** A **greedy multi-path valid verification algorithm** forms a global ranking on paths in $\mathcal{V}^L$, and always selects the highest-ranked drafted path in the path selection rule $\Gamma$ (Lemma 4.4).

In the context of the multi-path LP, each greedy multi-path valid verification algorithms can be viewed as the output of a greedy polymatroid algorithm (Schrijver et al., 2003) on the submodular feasibility constraints in Theorem 4.3, with the algorithm-selected order being the same as the global ranking. Due to space constraints, we expand on this connection in detail in Appendix H.

If the global ordering of paths is $\mathcal{V}^L = \{P_1, \ldots, P_M\}$, then one selects $P_i$ from $\Gamma$ if and only if all $K$ drafted paths lie in $\{P_1, \ldots, P_i\}$, but do not all lie $\{P_1, \ldots, P_{i-1}\}$. Thus, we can explicitly write:

$$q^\Gamma(P_i) = \left(\sum_{j=1}^{i} q(P_j)\right)^K - \left(\sum_{j=1}^{i-1} q(P_j)\right)^K. \tag{19}$$

Again, computing this for arbitrary global orderings may require off-path probability values. However, when the paths are ordered through a tree-based rule, only on-path values are needed. Tree-based rules create a global ranking by generating local orderings at all prefixes in $\mathcal{V}^{\leq L}$, and then combine them to induce a lexicographic ordering over all length-$L$ paths.

**Definition 5.2.** A global ranking of paths in $\mathcal{V}^L$ is **tree-based** if the ranking can be obtained as follows. Define an injective function $\pi_{a_{1:i}} : \mathcal{V} \to \mathbb{R}$ at each $a_{1:i} \in \mathcal{V}^{\leq L}$, only using the values $p(\cdot|a_{1:i}), q(\cdot|a_{1:i})$. Then, assign to each path $a_{1:L} \in \mathcal{V}^L$ the $L$-tuple $O(a_{1:L}) = (\pi_{a_{1:i-1}}(a_i))_{i=1}^{L}$. Finally, rank the paths $a_{1:L} \in \mathcal{V}^L$ in increasing lexicographic order of $O(a_{1:L})$.

---

**Algorithm 1:** Greedy Multi-Path Block Verification

---

**Input:** Draft blocks $X_{1:L}^{(1)}, \ldots, X_{1:L}^{(K)}$, target and draft probabilities $p\left(\cdot | X_{1:i}^{(k)}\right), q\left(\cdot | X_{1:i}^{(k)}\right)$

**1 for** $k = 1, \ldots, K$ **do**

**2** $\quad$ Define $k$th path ranking $O_k$ by $\left[p\left(X_i^{(k)} | X_{1:i-1}^{(k)}\right) / q(X_i^{(k)} | X_{1:i-1}^{(k)})\right]_{i=1}^{L}$

**3** Select $k_0$ with maximal ranking $O_{k_0}$ by lexicographic ordering

**4 for** $i = 1, \ldots, L$ **do**

**5** $\quad$ Compute $q^{\Gamma_g}\left(\cdot | X_{1:i}^{(k_0)}\right)$ from $p\left(\cdot | X_{1:i}^{(k_0)}\right), q\left(\cdot | X_{1:i}^{(k_0)}\right)$ using Equation (23)

**6** Run block verification on $X_{1:L}^{(k_0)}$ with target and draft probabilities $p\left(\cdot | X_{1:i}^{(k_0)}\right), q^{\Gamma_g}\left(\cdot | X_{1:i}^{(k_0)}\right)$

---

Now, the question remains of how to choose the local orderings $\pi_{a_{1:i}}$. Our key observation is that from Equation (19), and the convexity of $X \mapsto X^K$, we get

$$\frac{q^\Gamma(P_1)}{q(P_1)} \le \frac{q^\Gamma(P_2)}{q(P_2)} \le \ldots \le \frac{q^\Gamma(P_M)}{q(P_M)}. \tag{20}$$

That is, $q^\Gamma/q$ increases along the global ranking. Hence, to make $q^\Gamma/p$ heuristically close to one, we would also like $p/q$ to increase along the global ranking. Again, enforcing this strict global requirement is difficult, as it requires knowledge of the full joint $p_L$ and $q_L$. Therefore, we further relax this to a tree-based ranking, by making each local ordering follow $p/q$ in increasing order:

$$\pi_{a_{1:i}}(x) = \frac{p(x|a_{1:i})}{q(x|a_{1:i})}. \tag{21}$$

We denote the path selection rule induced by these $\pi_{a_{1:i}}$ as $\Gamma_g$. Now, we can efficiently compute each $q^{\Gamma_g}(a_{1:i})$ using Equation (19). By definition of the lexicographic ordering, the second sum consists of all paths $b_{1:L}$ where for some $0 \le j \le i - 1$, we have $b_{1:j} = a_{1:j}$ and $\pi_{a_{1:j}}(b_{j+1}) < \pi_{a_{1:j}}(a_{j+1})$. The first sum contains all these paths, as well as all paths $b_{1:L}$ with $b_{1:i} = a_{1:i}$. This leads to the following closed form expression for $q^{\Gamma_g}$:

$$q^{\Gamma_g}(a_{1:i}) = \left(q(a_{1:i}) + \sum_{j=0}^{i-1} q(a_{1:j}) \sum_{\pi_{a_{1:j}}(t) < \pi_{a_{1:j}}(a_{j+1})} q(t)\right)^K \tag{22}$$

$$- \left(\sum_{j=0}^{i-1} q(a_{1:j}) \sum_{\pi_{a_{1:j}}(t) < \pi_{a_{1:j}}(a_{j+1})} q(t)\right)^K. \tag{23}$$

In fact, given a fixed $a_{1:i-1}$, we can efficiently compute all $q^{\Gamma_g}(a_{1:i})$ for $a_i \in \mathcal{V}$. Using the above expression, these $|\mathcal{V}|$ quantities have the exact same terms except at $q(a_{1:i})$ and the $j = i - 1$ term in the summation. We can compute the former over all $a_i \in \mathcal{V}$ in $O(|\mathcal{V}|)$ time by multiplying the conditional distribution $q(\cdot|a_{1:i-1})$ by $q(a_{1:i-1})$. The latter is a sum of $q(t)$ over all $t \in \mathcal{V}$ with $\pi_{a_{1:i-1}}(t) < \pi_{a_{1:i-1}}(a_i)$, so we can use a cumulative sum over the ordering on $\mathcal{V}$ induced by $\pi_{a_{1:i-1}}$. From these $|\mathcal{V}|$ values, we can compute conditional probabilities $q^{\Gamma_g}(\cdot|a_{1:i})$ using Equation (14).

Finally, now that we have an explicit path selection rule $\Gamma_g$ and corresponding skewed draft $q^{\Gamma_g}$ values, both of which are efficient to compute and only depend on on-path probabilities, we can follow Lemma 4.4 and use block verification with $q^{\Gamma_g}$ for the valid single-path verification algorithm. This results in **greedy multi-path block verification (GBV)**, which we explicitly enumerate in Algorithm 1. In the case $K = 1$, this is equivalent to single-path block verification with draft distribution $q$ and target distribution $p$, because $\Gamma_g$ only has one path to select, and thus $q^{\Gamma_g} = q$.

## 6 EXPERIMENTS

We now test GBV for $K = 1, 2, 3, 4$ paths[2]. As mentioned in Section 5, the case $K = 1$ is equivalent to single-path block verification from Sun et al. (2024b), so it is our baseline approach. For all

---

[2]We release our code here with implementations of other multi-path methods.

experiments, we select a block length of $L = 8$. We evaluate single-batch sampling from two target-draft model pairs: OPT 6.7B/350M and OPT 6.7B/125M (Zhang et al., 2022). All temperatures are set to 1.0. For each pair, we evaluate our algorithm on three datasets: GSM8K, HumanEval, and MATH500 (Chen et al., 2021; Hendrycks et al., 2021; Cobbe et al., 2021). We take 500 random problems from the test split of each dataset (HumanEval only has 164). We run our experiments on a Paperspace machine with an A100-80GB and an Intel Xeon Gold 6342 CPU (12 cores, 2.80 GHz). We measure *block efficiency* as the number of generated tokens per call to the target model (higher is better), and *walltime* as average milliseconds per token generated (lower is better).

| Model pair | Dataset | Block efficiency (tokens/$M_p$-call) | | | |
|---|---|---|---|---|---|
| | | $K = 1$ | $K = 2$ | $K = 3$ | $K = 4$ |
| OPT 6.7B/350M | GSM8K | 3.294 | 3.827 | 3.867 | **3.884** |
| | HumanEval | 3.157 | 3.538 | 3.809 | **3.863** |
| | MATH500 | 3.227 | 3.721 | 3.856 | **3.896** |
| OPT 6.7B/125M | GSM8K | 2.937 | 3.407 | 3.526 | **3.562** |
| | HumanEval | 2.653 | 3.023 | 3.233 | **3.503** |
| | MATH500 | 2.814 | 3.274 | 3.392 | **3.493** |

Table 1: We compute the block efficiency (tokens per target model call) for decoding with GBV, for target-draft model pairs OPT 6.7B/350M and OPT 6.7B/125M on up to 500 prompts from each of the datasets GSM8K, HumanEval, and MATH500. Larger numbers are better. In all settings, efficiency improves as $K$ increases from 1 (block verification baseline) to 4.

Our block efficiency results are shown in Table 1. Across all model pairs and datasets, increasing the number of draft paths $K$ monotonically improves block efficiency. From $K = 1$ to $K = 4$, block efficiency gains range from 17.91% to 32.04%, with an average 23.08% gain across all six model pair and dataset combinations. However, we also observe diminishing returns for higher $K$: on average, there is a 14.98% increase from $K = 1$ to $K = 2$, a 4.40% increase from $K = 2$ to $K = 3$, and a 2.54% increase from $K = 3$ to $K = 4$. Thus, while GBV significantly improves block efficiency, the most impactful gains come in the $K = 2$ setting.

| Model pair | Dataset | Walltime (ms/token) | | | |
|---|---|---|---|---|---|
| | | $K = 1$ | $K = 2$ | $K = 3$ | $K = 4$ |
| OPT 6.7B/350M | GSM8K | 44.482 | 39.592 | **38.862** | 48.473 |
| | HumanEval | 46.955 | 43.373 | **41.017** | 50.656 |
| | MATH500 | 45.851 | 40.473 | **38.889** | 48.440 |
| OPT 6.7B/125M | GSM8K | 34.479 | 30.862 | **30.109** | 39.831 |
| | HumanEval | 38.824 | 35.648 | **33.627** | 41.297 |
| | MATH500 | 36.022 | 32.166 | **31.163** | 40.329 |

Table 2: We compute the walltimes (ms/token) for decoding with GBV, for target-draft model pairs OPT 6.7B/350M and OPT 6.7B/125M on up to 500 prompts from each of the datasets GSM8K, HumanEval, and MATH500. Smaller numbers are better. In all settings, walltimes improve as $K$ increases from 1 (block verification baseline) to 3, but drop off at $K = 4$.

While block efficiency improvements are significant, they do not perfectly align with our walltime results in Table 2. Here, gains are no longer monotonic from $K = 1$ to $K = 4$. There are significant improvements from $K = 1$ to $K = 3$, with an average reduction in walltime by 13.34% across all settings. For OPT 6.7B/350M and MATH500, this even reaches a 15.19% reduction. However, $K = 4$ always performs worse than the baseline $K = 1$, with an average increase of 9.39% in walltime. This is due to the increased computation cost of performing the batched target forward pass over $K = 4$ paths (see Section 2.2), which negates block efficiency gains.

We also analyze decoding efficiency across datasets. The relative gains in block efficiency from $K = 1$ to $K = 4$ are highest for HumanEval (27.20% average across model pairs), followed by MATH500 (22.43%) and GSM8K (19.60%). For walltimes, the best setting $K = 3$ reduces walltime

relative to the baseline $K = 1$ by 12.65% for GSM8K, 14.34% for MATH500, and 13.02% for HumanEval (averaged across model pairs). While HumanEval benefits most from extra paths in block efficiency, these do not directly translate to walltime gains.

Furthermore, we examine the impact of target and draft model sizes. Averaged across datasets, the larger draft (350M) achieves 13.72% higher block efficiency than the smaller draft (125M) for $K = 3$. However, the smaller draft achieves a 20.14% lower average walltime than the larger draft in the same setting. While a larger draft model can improve acceptance rates and block efficiency, this comes at the expense of increased latency in draft sequence generation.

For practical usage, we recommend the setting $L = 8, K = 3$. This achieves nearly all of the block efficiency gains of $K = 4$, without incurring a significant spike in latency due to batched target calls. In all settings, $K = 3$ presents the fastest walltime. In multi-batch settings, $K = 2$ is also a viable alternative, as it has around a 10% average walltime reduction compared to block verification. Our results demonstrate that GBV significantly reduces end-to-end latency in autoregressive decoding.

## 6.1 OTHER MODEL FAMILIES AND DATASETS

While $K = 4$ has the highest block efficiency in our OPT experiments, this is not universally true. In Appendix I, we widen our experimental coverage to include model families with much larger target models (Qwen-3 32B/0.6B and Llama-3 70B/8B) and non-academic datasets (MGSM and ToolBench). In all these settings, we find that block verification outperforms GBV with $K > 1$ in both block efficiency and walltime. We also provide a systems breakdown of relative draft and target pass costs and the KV-cache footprint to explain cost breakdowns for larger models. These results show how to best deploy GBV. For some families (OPT) $K = 3$ is optimal, whereas for others (Qwen-3 and Llama-3) the best choice is to revert to block verification, i.e. GBV with $K = 1$.

## 6.2 TEMPERATURE AND DRAFT LENGTH ABLATIONS

In Appendix J, we expand our Qwen-3 32B/0.6B experiments for GSM8K in Appendix I from temperature 1.0 target sampling to 0.2, 0.4, 0.6, and 0.8. We find that block verification performs worse and GBV with $K > 1$ performs better at lower temperatures. At temperature 0.2, GBV with $K = 3$ is over 0.6 tokens/s faster than block verification. Thus, even for model families where block verification is better at temperature one, GBV with $K > 1$ can be better at low temperatures.

In Appendix K, we perform ablations on the draft length $L$. We find that increasing our $L = 8$ setting to $L = 16, 24$ yields modest gains in block efficiency, but throughput degrades rapidly. Following a systems breakdown similar to Appendix I, we explain this occurs because longer block lengths push more relative work into the drafting bottleneck. We find the ideal choice of $L$ is somewhere around 8.

## 6.3 COMPARISON TO MULTI-PATH METHODS

Finally, we compare GBV to multi-path verification algorithms SpecTr (Sun et al., 2023), SpecInfer (Miao et al., 2024), and traversal verification (Weng et al., 2025). The rationale for using these methods along with detailed results and discussion are in Appendix L. Traversal verification beats GBV at temperature 1.0 by up to 1.0 tokens/s, GBV beats traversal verification at temperature 0.2 by up to 1.6 tokens/s, and other methods lag behind. In its best setting of $K = 2$, temperature 0.2, GBV achieves nearly 11.5 tokens/s, whereas no other method achieves over 10 tokens/s in *any* setting.

## 7 CONCLUSION

We developed a single-path information-agnostic linear program (LP) which encodes feasible speedups for valid verification algorithms in speculative sampling, and showed that block verification remains optimal even with access to off-path probabilities. We further extended our LP to the setting where multiple draft paths are generated. While this multi-path LP is not feasible to solve exactly, by approximating it with a class of greedy verification algorithms, we developed a generalization of block verification, called greedy multi-path block verification. This significantly improves decoding efficiency relative to block verification. Future work could further explore the class of greedy verification schemes, or explore alternative approximation to the multi-path LP.

## REPRODUCIBILITY STATEMENT

The majority of our work is theoretical, with proofs in Appendices A to G. While we do not release code, we provide enough detail in Section 5 to reproduce our greedy multi-path block verification algorithm. We also provide details around our machine setup, datasets, and model pairs in Section 6.

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

## A  PROOF OF MULTI-PATH INFORMATION-AGNOSTIC LP

In this section, we prove the multi-path information-agnostic LP in Theorem 4.3. Our proof begins with a key lemma that dictates when one can sample from one distribution given data from another, with the conditional distribution (often called the transport) to induce the former restricted to a given support. In our context, the restricted support represents the prefix-matching requirement, the distribution from which we are given data is induced by i.i.d. sampling paths from the draft model, and the distribution from which we would like to sample is that of the verification algorithm output.

**Lemma A.1** (Bipartite Transport Feasibility). *Let $G = (A \cup B, E \subseteq A \times B)$ be a bipartite graph, with probability distributions $a(\cdot)$ and $b(\cdot)$ over $A$ and $B$, respectively. Then the following conditions are equivalent, where $N(\cdot)$ denotes neighborhoods:*

1. *There exists a joint distribution $\pi(\cdot, \cdot)$ over $A \times B$ with marginal distributions $a(\cdot)$ and $b(\cdot)$, such that $\pi(x, y) = 0$ for all $(x, y) \notin E$.*

2. *For any $S \subseteq A$, we have $\sum_{x \in S} a(x) \leq \sum_{y \in N(S)} b(y)$.*

*Proof.* Condition 1 is equivalent to the feasibility of the following LP in variables $\pi_{x,y} \geq 0$:

$$\sum_{x \in A} \pi_{x,y} = b(y) \quad \forall y \in B, \quad \sum_{y \in B} \pi_{x,y} = a(x) \quad \forall x \in A, \quad \pi_{x,y} = 0 \quad \forall (x, y) \notin E. \tag{24}$$

We can incorporate the zero equality condition into the sum equalities to turn this into:

$$\sum_{x \in N(y)} \pi_{x,y} = b(y) \quad \forall y \in B, \quad \sum_{y \in N(x)} \pi_{x,y} = a(x) \quad \forall x \in A. \tag{25}$$

Now, Gale's feasibility theorem for bipartite supply-demand networks (Gale, 1957) implies this LP is feasible in nonnegative variables if and only if

$$\sum_{x \in S} a(x) \leq \sum_{y \in N(S)} b(x) \quad \forall S \subseteq A, \quad \sum_{x \in A} a(x) = \sum_{y \in B} b(x). \tag{26}$$

The equality condition is automatically satisfied, since $a(\cdot), b(\cdot)$ are probability distributions, so both sides become one. Thus, Condition 1 is equivalent to Condition 2. $\square$

Using this, we can prove the following lemma, which dictates what distributions are feasible for a $K$-path verification algorithm $\Phi$ which satisfies prefix-matching. We do not enforce target-matching yet, so this algorithm is not necessarily valid. For any $K$-path verification algorithm $\Phi$, denote $R_\Phi(\cdot)$ as the distribution of the output of $\Phi$, which is supported over $\mathcal{V}^{\leq L+1} \setminus \mathcal{V}^0$, i.e. all nonempty paths of length $\leq L + 1$. We need the nonempty requirement as we must output at least one token. As we prove in Lemma A.2, a collection of subset-sum inequalities over a particular bipartite graph exactly describes the set of $R_\Phi$ which can be realized by some $\Phi$ satisfying prefix-matching.

**Lemma A.2** (Prefix-Matching). *Construct the bipartite graph $G$ with left vertices $V^{\leq L+1} \setminus \mathcal{V}^0$, right vertices $(\mathcal{V}^L)^K$, and edges $E$ consisting of all pairs*

$$\left(a_{1:i}, \left(a_{1:L}^{(1)}, \ldots, a_{1:L}^{(K)}\right)\right) \in \left(\mathcal{V}^{\leq L+1} \setminus \mathcal{V}^0\right) \times \left(\mathcal{V}^L\right)^K, \quad a_{1:i-1} \in \left\{a_{1:i-1}^{(1)}, \ldots, a_{1:i-1}^{(K)}\right\}. \tag{27}$$

*The values $R_\Phi(a_{1:i})$ for $a_{1:i} \in \mathcal{V}^{\leq L+1} \setminus \mathcal{V}^0$ are feasible for a $K$-path verification algorithm $\Phi$ that satisfies prefix-matching if and only if they are nonnegative, sum to one, and*

$$\sum_{a_{1:i} \in S} R_\Phi(a_{1:i}) \leq \sum_{\left(a_{1:L}^{(1)}, \ldots, a_{1:L}^{(K)}\right) \in N(S)} \prod_{k=1}^{K} q\left(a_{1:L}^{(k)}\right) \quad \forall S \subseteq V^{\leq L+1} \setminus \mathcal{V}^0, \tag{28}$$

*where $N(\cdot)$ denotes a neighborhood in $G$.*

*Proof.* The sum to one equality and nonnegative requirements are equivalent to requiring that $R_\Phi$ is a valid probability distribution. Now, given that $R_\Phi$ is a valid probability distribution, we observe that the prefix-matching requirement is equivalent to the following: given $K$ i.i.d. sampled paths

$$X_{1:L}^{(1)}, \ldots, X_{1:L}^{(K)} \sim q(\cdot), \tag{29}$$

there is a conditional distribution (randomized rule) $\pi(\cdot | X_{1:L}^{(1)}, \ldots, X_{1:L}^{(K)})$ over $\mathcal{V}^{\leq L+1} \setminus \mathcal{V}^0$ which is supported only over prefixes of these paths plus an additional token, such that sampling from $\pi$ results in the distribution $R_\Phi$. Given the construction of the graph $G$, we can alternatively express the restricted support requirement on $\pi(\cdot|\cdot)$ as $\pi(v_l|v_r) = 0$ for all left vertices $v_l$ and right vertices $v_r$ with $(v_l, v_r) \notin E$. Thus, by turning this conditional distribution into a joint distribution, as $R_\Phi$ is a distribution over left vertices and i.i.d. sampling from $q(\cdot)$ is a distribution over right vertices, Lemma A.1 directly applies to show that prefix-matching is equivalent to

$$\sum_{a_{1:i} \in S} R_\Phi(a_{1:i}) \leq \sum_{\left(a_{1:L}^{(1)}, \ldots, a_{1:L}^{(K)}\right) \in N(S)} \prod_{k=1}^{K} q\left(a_{1:L}^{(k)}\right) \quad \forall S \subseteq V^{\leq L+1} \setminus \mathcal{V}^0. \tag{30}$$

The product term is the probability of sampling $K$ paths i.i.d from $q$. This completes the proof. □

Now, we move onto the target-matching requirement. Again, we can establish a necessary and sufficient condition for a $K$-path verification algorithm $\Phi$ to satisfy target-matching terms of the $R_\Phi$ values. Now, the condition is path-wise summation requirement.

**Lemma A.3** (Target-Matching). *The values $R_\Phi(a_{1:i})$ for $a_{1:i} \in \mathcal{V}^{\leq L+1} \setminus \mathcal{V}^0$ are feasible for a $K$-path verification algorithm $\Phi$ that satisfies prefix-matching if and only if they are nonnegative, sum to one, and*

$$\sum_{i=0}^{L} \frac{R_\Phi(a_{1:i+1})}{p(a_{1:i+1})} = 1 \quad \forall a_{1:L+1} \in \mathcal{V}^{L+1}. \tag{31}$$

*Proof.* Target-matching asserts that sampling from $p_{L+1}(\cdot)$ is equivalent to sampling the output $(X_{1:\tau}, Y) \sim R_\Phi$ of the verification algorithm and then sampling an additional $L - \tau$ tokens from $Z_{1:L-\tau} \sim p_{L-\tau}(\cdot|X_{1:\tau}, Y)$. For a given path $a_{1:L+1} \in \mathcal{V}^{L+1}$, the probability of sampling this path from the former method is $p_{L+1}(a_{1:L+1})$. The probability of sampling it from the latter method is

$$\mathbb{P}(X_{1:\tau} = a_{1:\tau}, Y = a_{\tau+1}, Z_{1:L-\tau} = a_{\tau+2:L+1}) \tag{32}$$

$$= \sum_{i=0}^{L} \mathbb{P}(\tau = i, X_{1:\tau} = a_{1:\tau}, Y = a_{\tau+1}, Z_{1:L-\tau} = a_{\tau+2:L+1}) \tag{33}$$

$$= \sum_{i=0}^{L} \mathbb{P}(\tau = i, X_{1:\tau} = a_{1:i}, Y = a_{i+1}, Z_{1:L-\tau} = a_{i+2:L+1}) \tag{34}$$

$$= \sum_{i=0}^{L} \mathbb{P}((X_{1:\tau}, Y) = a_{1:i+1})p(a_{i+2:L+1}|a_{1:i+1}) \tag{35}$$

$$= \sum_{i=0}^{L} R_\Phi(a_{1:i+1}) \cdot \frac{p(a_{1:L+1})}{p(a_{1:i+1})}. \tag{36}$$

Setting these two equal for all paths $a_{1:L+1}$ and dividing out the $p(a_{1:L+1}) = p_{L+1}(a_{1:L+1})$ term gives the desired condition. □

By combining the prefix-matching and target-matching lemmas, we can now write down the multi-path LP, in a form which is slightly more complicated than in Theorem 4.3, and is in terms of variables $R_\Phi$ rather than the node budgets $D_\Phi$. We will use the notion of a parent set and minimal set.

**Definition A.4.** For each $S \subseteq V^{\leq L+1} \setminus \mathcal{V}^0$, the **parent set** $\mathrm{par}(S) \subseteq \mathcal{V}^{\leq L}$ of $S$ consists of all $a_{1:i} \in \mathcal{V}^{\leq L}$ where $a_{1:i+1} \in S$ for some $a_{i+1} \in \mathcal{V}$. For each $S \subseteq V^{\leq L}$, the **minimal set** $\mathrm{min}(S) \subseteq S$ of $S$ consists of all $a_{1:i} \in S$ where no proper prefix of $a_{1:i}$ appears in $S$, i.e. $a_{1:0}, \ldots, a_{1:i-1} \notin S$.

Note that an antichain (Definition 4.2) is precisely a subset $S \subseteq \mathcal{V}^{\leq L}$ where $\min(S) = S$, and any minimal set $\min(S)$ is also an antichain because $\min(\min(S)) = \min(S)$. We will use parent and minimal sets to remove redundant inequalities in the Lemma A.2.

**Lemma A.5** (Unsimplified Multi-Path LP). *The values $R_\Phi(a_{1:i})$ for $a_{1:i} \in \mathcal{V}^{\leq L+1} \setminus \mathcal{V}^0$ are feasible for a valid $K$-path verification algorithm $\Phi$ if and only if they are nonnegative, sum to one, and are feasible in*

$$\max \sum_{i=1}^{L+1} \sum_{a_{1:i} \in \mathcal{V}^i} i R_\Phi(a_{1:i}), \tag{37}$$

$$s.t. \quad \sum_{i=0}^{L} \frac{R_\Phi(a_{1:i+1})}{p(a_{1:i+1})} = 1 \quad \forall a_{1:L+1} \in \mathcal{V}^{L+1}, \tag{38}$$

$$\sum_{a_{1:i} \in T} \sum_{j=i+1}^{L+1} \sum_{a_{i+1:j} \in \mathcal{V}^{j-i}} R_\Phi(a_{1:j}) \leq 1 - \left(1 - \sum_{a_{1:i} \in T} q(a_{1:i})\right)^K \quad \forall T \in \mathcal{A}(\mathcal{V}^{\leq L}). \tag{39}$$

*Furthermore, the objective value is precisely the block efficiency $\mathbb{E}[\tau + 1]$ for $\Phi$.*

*Proof.* First, fix a subset $S \subseteq \mathcal{V}^{\leq L+1} \setminus \mathcal{V}^0$. The neighborhood $N(S) \subseteq (\mathcal{V}^L)^K$ induced by the graph $G$ in Lemma A.2 is the set of $K$-tuples of length-$L$ paths where at least one path has a prefix in $\mathrm{par}(S)$. So, the complement of $N(S)$ consists of all $K$-tuples of length-$L$ paths where no path has a prefix in $\mathrm{par}(S)$. Now, note that a path has a prefix in $\mathrm{par}(S)$ if and only if it has a prefix in $\min(\mathrm{par}(S))$. Because $\min(\mathrm{par}(S))$ is an antichain (i.e. no two distinct elements in it are a prefix of the same path), disjointness shows that the probability that a length $L$-path sampled from $q(\cdot)$ has a prefix in $\min(\mathrm{par}(S))$ is $\sum_{a_{1:i} \in \min(\mathrm{par}(S))} q(a_{1:i})$. Using these observations, we simplify:

$$\sum_{\left(a_{1:L}^{(1)}, \ldots, a_{1:L}^{(K)}\right) \in N(S)} \prod_{k=1}^{K} q\left(a_{1:L}^{(k)}\right) = 1 - \sum_{\left(a_{1:L}^{(1)}, \ldots, a_{1:L}^{(K)}\right) \notin N(S)} \prod_{k=1}^{K} q\left(a_{1:L}^{(k)}\right) \tag{40}$$

$$= 1 - \left(1 - \sum_{a_{1:i} \in \min(\mathrm{par}(S))} q(a_{1:i})\right)^K. \tag{41}$$

Thus, the prefix-matching inequalities from Lemma A.2 become:

$$\sum_{a_{1:i} \in S} R_\Phi(a_{1:i}) \leq 1 - \left(1 - \sum_{a_{1:i} \in \min(\mathrm{par}(S))} q(a_{1:i})\right)^K \quad \forall S \subseteq V^{\leq L+1} \setminus \mathcal{V}^0. \tag{42}$$

We now make the key observation that many of these inequalities are redundant. For any $a_{1:i}$ with a proper (i.e. not full) prefix in $\min(\mathrm{par}(S))$, we can add $a_{1:i}$ to $S$ without changing $\min(\mathrm{par}(S))$, keeping the right hand side constant. This also increases the left hand side (since $R_\Phi$ is nonnegative). Thus, the strictest of these inequalities are precisely those where $S$ includes all paths with a proper prefix in $\min(\mathrm{par}(S))$. For a fixed $T = \min(\mathrm{par}(S))$, this makes $S$ the set of all $a_{1:j}$ where $a_{1:i} \in T$ for some $i < j$. Since all antichains are achievable as some $\min(\mathrm{par}(S))$, this turns the above collection of inequalities into

$$\sum_{a_{1:i} \in T} \sum_{j=i+1}^{L+1} \sum_{a_{i+1:j} \in \mathcal{V}^{\leq j-i}} R_\Phi(a_{1:j}) \leq 1 - \left(1 - \sum_{a_{1:i} \in T} q(a_{1:i})\right)^K \quad \forall T \in \mathcal{A}(\mathcal{V}^{\leq L}). \tag{43}$$

This covers the prefix-matching. The target-matching requirement remains the same as in Lemma A.3. Thus, all that remains is to show the objective values is the block efficiency. This holds as $R_\Phi(a_{1:i})$ is the probability that $\Phi$ outputs $a_{1:i}$, and the total number of tokens generated with this output is $i$, so summing over all possible outputs $a_{1:i} \in \mathcal{V}^{\leq L+1} \setminus \mathcal{V}^0$ gives the desired result. $\square$

Finally, we can prove the multi-path LP in Theorem 4.3. The only remaining step to derive this is to turn the variables $R_\Phi$ into variables $D_\Phi$ representing node budgets.

*Proof.* By the definition of $D_\Phi$ in Definition 3.2, we see that

$$D_\Phi(a_{1:i}) = 1 - \sum_{j=1}^{i} \frac{R_\Phi(a_{1:i})}{p(a_{1:i})} \quad \forall a_{1:i} \in \mathcal{V}^{\leq L+1}. \tag{44}$$

The target-matching condition simply implies that each $D_\Phi(a_{1:L+1}) = 0$. Now, observe that

$$R_\Phi(a_{1:i}) = D_\Phi(a_{1:i-1})p(a_{1:i}) - D_\Phi(a_{1:i})p(a_{1:i}) \quad \forall a_{1:i} \in \mathcal{V}^{\leq L}. \tag{45}$$

for all. Thus, incorporating the nonnegativity constraint on $R_\Phi$, we get

$$1 = D_\Phi(a_{1:0}) \geq D_\Phi(a_{1:1}) \geq \ldots \geq D_\Phi(a_{1:L}) \geq D_\Phi(a_{1:L+1}) = 0 \quad \forall a_{1:L+1} \in \mathcal{V}^{L+1}. \tag{46}$$

Next, observe that for any $a_{1:i} \in \mathcal{V}^{\leq L}$, we have the following telescoping identity:

$$\sum_{j=i+1}^{L+1} \sum_{a_{i+1:j} \in \mathcal{V}^{j-i}} R_\Phi(a_{1:j}) \tag{47}$$

$$= \sum_{j=i+1}^{L+1} \sum_{a_{i+1:j} \in \mathcal{V}^{j-i}} D_\Phi(a_{1:j-1})p(a_{1:j}) - \sum_{j=i+1}^{L+1} \sum_{a_{i+1:j} \in \mathcal{V}^{j-i}} D_\Phi(a_{1:j})p(a_{1:j}) \tag{48}$$

$$= \sum_{j=i}^{L} \sum_{a_{i+1:j+1} \in \mathcal{V}^{j+1-i}} D_\Phi(a_{1:j})p(a_{1:j+1}) - \sum_{j=i+1}^{L+1} \sum_{a_{i+1:j} \in \mathcal{V}^{j-i}} D_\Phi(a_{1:j})p(a_{1:j}) \tag{49}$$

$$= D_\Phi(a_{1:i})p(a_{1:i}) + \sum_{j=i+1}^{L} \sum_{a_{i+1:j+1} \in \mathcal{V}^{j+1-i}} D_\Phi(a_{1:j})p(a_{1:j+1}) \tag{50}$$

$$- \sum_{j=i+1}^{L} \sum_{a_{i+1:j} \in \mathcal{V}^{j-i}} D_\Phi(a_{1:j})p(a_{1:j}) - 0 \tag{51}$$

$$= D_\Phi(a_{1:i})p(a_{1:i}) + \sum_{j=i+1}^{L} D_\Phi(a_{1:j}) \left( \sum_{a_{i+1:j+1} \in \mathcal{V}^{j+1-i}} p(a_{1:j+1}) - \sum_{a_{i+1:j} \in \mathcal{V}^{j-i}} p(a_{1:j}) \right) \tag{52}$$

$$= D_\Phi(a_{1:i})p(a_{1:i}) + \sum_{j=i+1}^{L} D_\Phi(a_{1:j}) \left( p(a_{1:i}) - p(a_{1:i}) \right) = D_\Phi(a_{1:i})p(a_{1:i}). \tag{53}$$

This means the condition that all $R_\Phi$ sum to one is automatically satisfied by applying this identity for $i = 0$. This also reduces the antichain condition in the unsimplified multi-path LP to

$$\sum_{a_{1:i} \in T} D_\Phi(a_{1:i})p(a_{1:i}) \leq 1 - \left( 1 - \sum_{a_{1:i} \in T} q(a_{1:i}) \right)^K \quad \forall T \in \mathcal{A}(\mathcal{V}^{\leq L}). \tag{54}$$

This covers all conditions, so all that remains is to show that the objective reduces to the desired expression. This holds by interchanging the order of summation, and using the telescoping identity:

$$\sum_{i=1}^{L+1} \sum_{a_{1:i} \in \mathcal{V}^i} i R_\Phi(a_{1:i}) = \sum_{i=1}^{L+1} \sum_{j=0}^{i-1} \sum_{a_{1:i} \in \mathcal{V}^i} R_\Phi(a_{1:i}) \tag{55}$$

$$= \sum_{j=0}^{L} \sum_{i=j+1}^{L+1} \sum_{a_{1:i} \in \mathcal{V}^i} R_\Phi(a_{1:i}) \tag{56}$$

$$= \sum_{j=0}^{L} \sum_{i=j+1}^{L+1} \sum_{a_{1:j} \in \mathcal{V}^j} \sum_{a_{j+1:i} \in \mathcal{V}^{j-i}} R_\Phi(a_{1:i}) \tag{57}$$

$$= \sum_{j=0}^{L} \sum_{a_{1:j} \in \mathcal{V}^j} \sum_{i=j+1}^{L+1} \sum_{a_{j+1:i} \in \mathcal{V}^{j-i}} R_\Phi(a_{1:i}) = \sum_{a_{1:j} \in \mathcal{V}^{\leq L}} D_\Phi(a_{1:j})p(a_{1:j}). \tag{58}$$

This completes the reduction to the desired form for the multi-path LP. $\qquad\square$

## B  PROOF OF SINGLE-PATH INFORMATION-AGNOSTIC LP

Using the multi-path in Theorem 4.3, it is easy to prove the single-path LP in Theorem 3.3 by reducing it to the case $K = 1$. This works because a valid single-path verification algorithm is precisely the same as a valid $K$-path verification algorithm in the case $K = 1$.

*Proof.* It suffices to show that when $K = 1$, the multi-path information-agnostic LP from Theorem 4.3 reduces to the single-path LP. Indeed, the antichain condition becomes

$$\sum_{a_{1:i} \in T} D_\Phi(a_{1:i}) p(a_{1:i}) \leq \sum_{a_{1:i} \in T} q(a_{1:i}) \quad \forall T \in \mathcal{A}(\mathcal{V}^{\leq L}). \tag{59}$$

This forces the pointwise conditions $D_\Phi(a_{1:i}) p(a_{1:i}) \leq q(a_{1:i})$, since by taking the singleton antichains $T = \{a_{1:i}\}$. Furthermore, these pointwise conditions imply the antichain conditions through summation. All other conditions and the objective function remain the same. Hence, we obtain the desired single-path LP. □

## C  PROOF OF BLOCK VERIFICATION OPTIMALITY

Using the single-path LP, we can prove that block verification is optimal in the class of valid single-path verification algorithms (Theorem 3.4). We will use results from Sun et al. (2024b). The idea is to show that $D_{\Phi_{BV}}$ for the single-path block verification algorithm $\Phi_{BV}$ are tight in the single-path LP.

*Proof.* From Lemma 4 in Appendix B of Sun et al. (2024b), we have for each path $a_{1:i} \in \mathcal{V}^{\leq L}$ that

$$\mathbb{P}(\tau \geq i | X_{1:i} = a_{1:i}) = w(a_{1:i}) \tag{60}$$

under $\Phi_{BV}$, where weights $w$ are defined in Equation (3). Now, because $\Phi_{BV}$ is a valid verification algorithm, we can use the telescoping identity from the proof of Theorem 4.3 at the end of Appendix A (with $R_{\Phi_{BV}}$ defined similarly as the distribution of the output of $\Phi_{BV}$) to show

$$D_{\Phi_{BV}}(a_{1:i}) p(a_{1:i}) = \sum_{j=i+1}^{L+1} \sum_{a_{i+1:j} \in \mathcal{V}^{j-i}} R_{\Phi_{BV}}(a_{1:j}) \tag{61}$$

$$= \sum_{j=i+1}^{L+1} \sum_{a_{i+1:j} \in \mathcal{V}^{j-i}} \mathbb{P}(\tau = j - 1, X_{1:j-1} = a_{1:j-1}, Y = a_j) \tag{62}$$

$$= \sum_{j=i+1}^{L+1} \mathbb{P}(\tau = j - 1, X_{1:i} = a_{1:i}) = \mathbb{P}(\tau \geq i, X_{1:i} = a_{1:i}) \tag{63}$$

$$= \mathbb{P}(\tau \geq i | X_{1:i} = a_{1:i}) \mathbb{P}(X_{1:i} = a_{1:i}) = w(a_{1:i}) q(a_{1:i}). \tag{64}$$

Thus, to show block verification is optimal, it suffices to show that for any valid single-path verification algorithm $\Phi$ and path $a_{1:i} \in \mathcal{V}^{\leq L}$, we have $D_\Phi(a_{1:i}) p(a_{1:i}) \leq w(a_{1:i}) q(a_{1:i})$, as then the objective value for $\Phi$ (the block efficiency, i.e. the sum of these $D_\Phi p$ terms) cannot exceed the objective value for $\Phi_{BV}$ (the sum of the $D_{\Phi_{BV}} p$ terms) in the single-path LP. The proof is by induction on $i$. For the base case $i = 0$, this is clear as $w(a_{1:0}) q(a_{1:0}) = 1 = D_\Phi(a_{1:0}) p(a_{1:0})$. Now, suppose this statement holds for $i$. To prove it for $i + 1$, we use the single-path conditions to obtain

$$D_\Phi(a_{1:i+1}) p(a_{1:i+1}) \leq D_\Phi(a_{1:i}) p(a_{1:i+1}) \tag{65}$$

$$= D_\Phi(a_{1:i}) p(a_{1:i}) p(a_{i+1} | a_{1:i}) \tag{66}$$

$$\leq w(a_{1:i}) q(a_{1:i}) p(a_{i+1} | a_{1:i}) \tag{67}$$

$$= q(a_{1:i+1}) \cdot \frac{p(a_{i+1} | a_{1:i})}{q(a_{i+1} | a_{1:i})} \cdot w(a_{1:i}) \tag{68}$$

$$\leq q(a_{1:i+1}) \cdot \min \left\{ 1, \frac{p(a_{i+1} | a_{1:i})}{q(a_{i+1} | a_{1:i})} \cdot w(a_{1:i}) \right\} = q(a_{1:i+1}) w(a_{1:i+1}). \tag{69}$$

The last inequality holds by using the pointwise $D_\Phi p \leq q$ bound from the single-path LP. This completes the induction, and thus the proof that block verification is optimal.

□

# D  PROOF OF PATH SELECTION DECOMPOSITION

In this section, we prove Lemma 4.4, which states that any valid $K$-path verification algorithm can be decomposed into a randomized path selection rule which chooses one of the $K$ paths, and then a valid single-path verification algorithm that operates on an skewed draft distribution.

*Proof.* Let $\Phi$ denote the valid verification algorithm. This takes in $K$ paths $P_1, \ldots, P_K$ i.i.d. sampled from $q(\cdot)$, and samples a nonempty path in $\mathcal{V}^{\leq L+1}$ using a conditional distribution $\pi(\cdot|P_1, \ldots, P_K)$, which can depend on off-path target and draft distribution values. This is constrained to only output $(X_{1:\tau}, Y)$ where $X_{1:\tau}$ is a prefix of one of $P_1, \ldots, P_K$ and $Y \in \mathcal{V}$. Thus, we can conditionally factorize $\pi$ into a prefix selection rule $\pi_1$ on prefixes of $P_1, \ldots, P_K$ conditioned over $P_1, \ldots, P_K$, and an additional token rule $\pi_2$ on $\mathcal{V}$ conditioned over $P_1, \ldots, P_K, X_{1:\tau}$:

$$\pi(X_{1:\tau}, Y|P_1, \ldots, P_K) = \pi_1(X_{1:\tau}|P_1, \ldots, P_K)\pi_2(Y|P_1, \ldots, P_K, X_{1,\tau}). \tag{70}$$

For each prefix $a_{1:i}$ of some path in $P_1, \ldots, P_K$, denote its multiplicity $m(a_{1:i}) \geq 1$ as the number of paths that contain it as a prefix. For example, $m(\emptyset) = K$. Then, we define a path selection rule $\pi_3$ on $[K]$ conditioned over $P_1, \ldots, P_K$, as:

$$\pi_3(k_0|P_1, \ldots, P_K) = \sum_{i=0}^{L} \pi_1((P_{k_0})_{1:i}|P_1, \ldots, P_K)m((P_{k_0})_{1:i})^{-1}. \tag{71}$$

To see this is a valid probability distribution, observe that for each distinct $a_{1:i}$ appearing as a prefix of one of $P_1, \ldots, P_K$, when we sum the above formula over all $k_0 \in [K]$, we have exactly $m(a_{1:i})$ terms in the sum above with $(P_{k_0})_{1:i} = a_{1:i}$, each of which are $\pi_1(a_{1:i}|P_1, \ldots, P_K)m(a_{1:i})^{-1}$. Thus, the terms containing $a_{1:i}$ cancel to $\pi_1(a_{1:i}|P_1, \ldots, P_K)$, and summing this over all *distinct* $a_{1:i}$ gives a result of 1, because $\pi_1$ is a valid distribution. Also, define a prefix selection rule $\pi_4$ on prefixes of $P_{k_0}$ conditioned over $P_1, \ldots, P_K, k_0 \in [K]$:

$$\pi_4((P_{k_0})_{1:i}|P_1, \ldots, P_K, k_0) = \frac{\pi_1((P_{k_0})_{1:i}|P_1, \ldots, P_K)m((P_{k_0})_{1:i})^{-1}}{\sum_{i=0}^{L} \pi_1((P_k)_{1:i}|P_1, \ldots, P_K)m((P_k)_{1:i})^{-1}}, \tag{72}$$

which is also a valid probability distribution. We see that when sampling $k_0 \sim \pi_3(\cdot|P_1, \ldots, P_k)$ and then $X_{1:\tau} \sim \pi_4(\cdot|P_1, \ldots, P_K, k_0)$, we have

$$\mathbb{P}(X_{1:\tau} = a_{1:i}|P_1, \ldots, P_K) \tag{73}$$

$$= \sum_{k_0 \in [K]} \pi_3(k_0|P_1, \ldots, P_k)\pi_4(a_{1:i}|P_1, \ldots, P_K, k_0) \tag{74}$$

$$= \sum_{k_0 \in [K], (P_{k_0})_{1:i} = a_{1:i}} \pi_3(k_0|P_1, \ldots, P_k)\pi_4(a_{1:i}|P_1, \ldots, P_K, k_0) \tag{75}$$

$$= \sum_{k_0 \in [K], (P_{k_0})_{1:i} = a_{1:i}} \pi_1(a_{1:i}|P_1, \ldots, P_K)m(a_{1:i})^{-1} \tag{76}$$

$$= m(a_{1:i})\pi_1(a_{1:i}|P_1, \ldots, P_K)m(a_{1:i})^{-1} = \pi_1(a_{1:i}|P_1, \ldots, P_K). \tag{77}$$

Thus, sampling from $\pi_3$ and then $\pi_4$ is equivalent to sampling from $\pi_1$, so sampling from $\pi$ is equivalent to sampling in the order $\pi_3, \pi_4, \pi_2$. Now, let $\pi_5$ be the distribution formed by sampling from $\pi_4$ and then $\pi_2$. Then sampling from $\pi$ is equivalent to sampling in the order $\pi_3, \pi_5$, where $\pi_3$ is a path selection rule conditioned on $P_1, \ldots, P_K$, and $\pi_5$ returns $(X_{1:\tau}, Y)$ conditioned only the selected path and $P_1, \ldots, P_K$. Now, denote $q^{\pi_3}$ as the true distribution of the path select by $\pi_3$. Then the output of $\Phi$ follows the distribution induced by sampling a *single* path from $q^{\pi_3}$, and sampling from $\pi_5$ conditioned only on that single path (the conditioning over $P_1, \ldots, P_K$ will cancel). To meet target-matching and prefix-matching, $\pi_5$ must correspond to a valid single-path verification algorithm with draft distribution $q^{\pi_3}$ and target distribution $p$. This completes the proof of the decomposition into a path selection rule $\pi_3$ and a valid single-path verification algorithm with draft values being the true distribution of the $\pi_3$-selected path. $\square$

## E    PROOF OF SKEWED DRAFT DISTRIBUTION FEASIBILITY

Here, we prove Lemma 4.5, which describes when a distribution can be realized as a skewed draft distribution $q^\Gamma$ from a given draft $q$ and path count $K$. The proof is a direct application of Lemma A.1.

*Proof.* Define the bipartite graph $G$ with left vertices $(\mathcal{V}^L)^K$, right vertices $\mathcal{V}^L$, and edges $E$ consisting of pairs $((P_1, \ldots, P_K), P_k)$ for all $P_1, \ldots, P_K \in \mathcal{V}^L$ and $k \in [K]$. Our distribution over left vertices is i.i.d. sampling from $q$, and our distribution over right vertices is $r$. We would like to find when there is a joint distribution on $(\mathcal{V}^L)^K \times \mathcal{V}^L$ supported on $E$, whose marginals are the left and right vertex distributions. Lemma A.1 shows such a distribution exists if and only if

$$\sum_{P \in S} r(P) \leq \sum_{(P_1, \ldots, P_K) \in N(S)} \prod_{k=1}^{K} q(P_k) \quad \forall S \subseteq \mathcal{V}^L. \tag{78}$$

Here, $N(\cdot)$ denotes neighborhoods in $G$. Thus, $N(S)$ consists of all $(P_1, \ldots, P_K)$ where some $P_k \in S$. This means the complement of $N(S)$ is precisely $(\mathcal{V}^L \setminus S)^K$, so the above becomes

$$\sum_{P \in S} r(P) \leq 1 - \left( \sum_{P \in \mathcal{V}^L \setminus S} q(P) \right)^K \quad \forall S \subseteq \mathcal{V}^L. \tag{79}$$

We are almost at our desired condition, except we need to show this holds for all antichains $S \in \mathcal{A}(\mathcal{V}^L)$. Indeed, for an antichain $S$, consider the set $P_S \subseteq \mathcal{V}^L$ of length-$L$ paths with a prefix in the antichain. No path in $P_S$ can have two distinct elements in $S$ as a prefix. Thus, the mass of $q$ over $S$ is the same as the mass of $q$ over $P_S$:

$$\sum_{a_{1:i} \in S} q(a_{1:i}) = \sum_{a_{1:i} \in S} \sum_{a_{i+1:L} \in \mathcal{V}^{L-i}} q(a_{1:L}) = \sum_{a_{1:L} \in P_S} q(a_{1:L}) \tag{80}$$

The same holds for $r$, because it is also a distribution over $\mathcal{V}^L$. Thus, the antichain conditions reduce to the conditions for $S \subseteq \mathcal{V}^L$ above, as desired. $\qquad \square$

## F    PROOF OF OPTIMAL MULTI-PATH ALGORITHM DESCRIPTION

Now, we prove Theorem 4.6, which describes the optimal multi-path valid verification algorithm as a solution to a nonlinear optimization problem.

*Proof.* The first part of the theorem is a direct consequence of the fact that block verification is the optimal single-path valid verification algorithm (Theorem 3.4), and any valid multi-path verification algorithm can be decomposed into path selection and valid single-path verification (Lemma 4.4). Now, we prove the second part of the theorem, using this description. Recall from the proof of Lemma 4.5 in Appendix E that a distribution $q^\Gamma$ can be realized from a given draft $q$ and path count $K$ with a path selection rule if and only if

$$q^\Gamma(a_{1:L}) \leq 1 - \left( 1 - \sum_{a_{1:L} \in T} q(a_{1:L}) \right)^K \quad \forall T \subseteq \mathcal{V}^L. \tag{81}$$

This covers the feasibility condition, so all that remains is to show the objective function in the theorem equals the block efficiency for block verification run on $q^\Gamma$. By using the expression for $D_{\Phi_{BV}} p$ in the proof of Theorem 3.4 in Appendix C, the objective value in the single-path LP (Theorem 3.3) simplifies to

$$\sum_{a_{1:i} \in \mathcal{V}^{\leq L}} D_{\Phi_{BV}}(a_{1:i}) p(a_{1:i}) = \sum_{a_{1:i} \in \mathcal{V}^{\leq L}} w(a_{1:i}) q^\Gamma(a_{1:i}), \tag{82}$$

where the weights $w$ are defined as in Equation (3), but with $q$ replaced by $q^\Gamma$. Thus, all that remains is to show that

$$w(a_{1:i}) q^\Gamma(a_{1:i}) = \min_{0 \leq k \leq i} p(a_{k+1:i} | a_{1:k}) q^\Gamma(a_{1:k}). \tag{83}$$

The proof is by induction on $i$. For $i = 0$, this is clear as both sides are one. Now, suppose this statement holds for $i$. To show it is true for $i + 1$, observe that

$$w(a_{1:i+1})q^\Gamma(a_{1:i+1}) = \min\left\{1, \frac{p(a_{i+1}|a_{1:i})}{q^\Gamma(a_{i+1}|a_{1:i})}w(a_{1:i})\right\}q^\Gamma(a_{1:i+1}) \tag{84}$$

$$= \min\left\{q^\Gamma(a_{1:i+1}), p(a_{i+1}|a_{1:i})q^\Gamma(a_{1:i})w(a_{1:i})\right\} \tag{85}$$

$$= \min\left\{q^\Gamma(a_{1:i+1}), p(a_{i+1}|a_{1:i}) \cdot \min_{0 \leq k \leq i} p(a_{k+1:i}|a_{1:k})q^\Gamma(a_{1:k})\right\} \tag{86}$$

$$= \min\left\{q^\Gamma(a_{1:i+1}), \min_{0 \leq k \leq i} p(a_{k+1:i+1}|a_{1:k})q^\Gamma(a_{1:k})\right\} \tag{87}$$

$$= \min_{0 \leq k \leq i+1} p(a_{k+1:i+1}|a_{1:k})q^\Gamma(a_{1:k}). \tag{88}$$

This completes the proof. $\qquad\square$

## G   PROOF OF OPTIMAL MULTI-PATH LOWER BOUND

We now prove Lemma 4.7, which lower bounds the objective value in Theorem 4.6.

*Proof.* Denote the minimum value of $q^\Gamma/p$ as

$$\epsilon = \min_{a_{1:i} \in \mathcal{V}^{\leq L}} \frac{q^\Gamma(a_{1:i})}{p(a_{1:i})}. \tag{89}$$

Then for each $a_{1:i} \in \mathcal{V}^{\leq L}$ and $0 \leq k \leq i$, we have the lower bound

$$p(a_{k+1:i}|a_{1:k})q^\Gamma(a_{1:k}) = \frac{q^\Gamma(a_{1:k})}{p(a_{1:k})} \cdot p(a_{1:i}) \geq \epsilon \cdot p(a_{1:i}). \tag{90}$$

Thus, we have the desired lower bound on the objective value:

$$\sum_{a_{1:i} \in \mathcal{V}^{\leq L}} \min_{0 \leq k \leq i} p(a_{k+1:i}|a_{1:k})q^\Gamma(a_{1:k}) \geq \sum_{a_{1:i} \in \mathcal{V}^{\leq L}} \epsilon \cdot p(a_{1:i}) = (L+1)\epsilon. \tag{91}$$

$\qquad\square$

## H   GREEDY POLYMATROID CONNECTION

Here, we describe the connection between greedy multi-path valid verification algorithms (Definition 5.1) and the greedy polymatrid algorithm. We start with Theorem 4.6, but only consider the feasibility condition:

$$q^\Gamma(a_{1:L}) \leq 1 - \left(1 - \sum_{a_{1:L} \in T} q(a_{1:L})\right)^K \quad \forall T \subseteq \mathcal{V}^L. \tag{92}$$

As previously mentioned, the right hand side is a submodular function $\psi(T)$ in $T$. Note that the inequality must also be an equality at $T = \mathcal{V}^L$ to ensure that $q^\Gamma$ is a probability distribution. Thus, to find *some* feasible $q^\Gamma$ satisfying these submodular constraints, we can use the greedy polymatroid algorithm from Schrijver et al. (2003), which aims to maximize the sum of $q^\Gamma$ terms given the above constraints. First, we fix any ordering $\{P_1, \ldots, P_M\}$ on $\mathcal{V}^L$. Then, a feasible solution is

$$q^\Gamma(P_i) = \psi(\{P_1, \ldots, P_i\}) - \psi(\{P_1, \ldots, P_{i-1}\}). \tag{93}$$

This is precisely what Equation (19) reduces to when the path selection rule $\Gamma$ is induced by the *reversed* global ordering $\{P_M, \ldots, P_1\}$. Thus, greedy multi-path valid verification algorithms are simply outputs of the greedy polymatroid algorithm for the multi-path LP feasibility conditions.

# I    OTHER MODEL FAMILIES AND DATASETS

Here, we expand our original evaluations for OPT-6.7B/350M/125M on GSM8K, HumanEval, and MATH500. We broaden our model family coverage to include model pairs Qwen-3 32B/0.6B and Llama-3 70B/8B, which allows us to test the efficacy of our approach on much larger target models. Due to memory constraints, the Llama-3 experiments were run on a Paperspace machine with 4xA6000 and an Intel Xeon Gold 5315Y CPU (32 cores, 3.20 GHz). We further expand the Qwen-3 dataset coverage to include 500 prompts from MGSM and ToolBench, to measure performance on diverse, non-academic tasks. Our block efficiency results for $L = 8$ and temperature 1.0 target sampling are shown in Table 3, and walltime numbers are in Table 4.

Interestingly, for Qwen-3, the single-path block verification baseline, i.e. $K = 1$, is most effective. On GSM8K, block efficiency is 4.32 tokens/$M_p$-call at $K = 1$, but decreases to 3.91, 3.57, and 3.43 for $K = 2, 3, 4$ respectively, while walltime increases from 79.6 to 92.8, 100.5, and 106.1 ms/token. Similar trends hold across HumanEval, MATH500, MGSM, and ToolBench. Likewise, for Llama-3, $K = 1$ again appears to be the optimal setting. Nevertheless, unlike Qwen-3, there can be meaningful gains at higher $K$. On GSM8K, the walltime decreases by nearly 10% from $K = 3$ to $K = 4$.

| Model pair | Dataset | Block efficiency (tokens/$M_p$-call) | | | |
|---|---|---|---|---|---|
| | | $K = 1$ | $K = 2$ | $K = 3$ | $K = 4$ |
| Qwen-3 32B/0.6B | GSM8K | **4.322** | 3.910 | 3.571 | 3.429 |
| | HumanEval | **4.204** | 3.423 | 3.331 | 2.960 |
| | MATH500 | **4.342** | 3.917 | 3.693 | 3.411 |
| | MGSM | **4.329** | 3.973 | 3.468 | 3.445 |
| | ToolBench | **3.090** | 2.706 | 2.791 | 2.681 |
| Llama-3 70B/8B | GSM8K | **4.943** | 4.430 | 4.271 | 4.625 |
| | HumanEval | **5.993** | 4.806 | 4.473 | 4.370 |

Table 3: We compute the block efficiency (tokens per target model call) for decoding with GBV, for target-draft model pairs Qwen-3 32B/0.6B and Llama-3 70B/8B on up to 500 prompts from each of the datasets GSM8K, HumanEval, and MATH500, MGSM, and ToolBench (only the first two for Llama-3). Larger numbers are better. In almost all settings, efficiency decreases monotonically as $K$ increases from 1 (block verification baseline) to 4.

| Model pair | Dataset | Walltime (ms/token) | | | |
|---|---|---|---|---|---|
| | | $K = 1$ | $K = 2$ | $K = 3$ | $K = 4$ |
| Qwen-3 32B/0.6B | GSM8K | **79.638** | 92.823 | 100.547 | 106.123 |
| | HumanEval | **84.166** | 106.051 | 109.215 | 124.554 |
| | MATH500 | **81.517** | 91.133 | 97.880 | 106.575 |
| | MGSM | **80.992** | 90.312 | 102.296 | 105.387 |
| | ToolBench | **113.563** | 132.477 | 128.405 | 133.973 |
| Llama-3 70B/8B | GSM8K | **114.123** | 133.188 | 136.272 | 123.545 |
| | HumanEval | **96.063** | 124.540 | 131.889 | 135.186 |

Table 4: We compute the walltimes (ms/token) for decoding with GBV, for target-draft model pairs Qwen-3 32B/0.6B and Llama-3 70B/8B on up to 500 prompts from each of the datasets GSM8K, HumanEval, and MATH500, MGSM, and ToolBench (only the first two for Llama-3). Smaller numbers are better. In almost all settings, walltime increases monotonically as $K$ increases from 1 (block verification baseline) to 4.

In our Llama-3 70B/8B experiments, we profiled draft tree size and saw that it grows roughly linearly with $K$, at around $9, 16, 24, 29$ for $K = 1, 2, 3, 4$. Across the $K$ sweep, the fraction of walltime spent in draft and target passes remains nearly constant (around 50% draft and 40% target), and the peak target and draft KV-cache footprints remain in a narrow band (target is around 80 MB on GSM8K and 110 MB on HumanEval, with the draft being around 40 MB for both).

This shows that increasing $K$ scales FLOPs and KV traffic in proportion to the larger tree, rather than changing the draft bottleneck. Even if block efficiency is high, end-to-end walltime degrades with $K$.

For Qwen-3 32B/0.6B, we performed the same profiling with similar results. Again, draft tree size grows approximately linearly with $K$. The peak KV-cache footprint also stays in a narrow band as $K$ increases, with the target ranging from roughly 60 to 120 MB across datasets and the draft from around 30 to 50 MB. Further, draft and target pass times also remain nearly constant relative to the walltime. However, compared to Llama-3, the key difference is that 70% of time is spent in drafting and only around 20% is in the batched target forward pass, with only small variations across $K$.

The target forward pass dominates the cost for Llama-3 significantly more than for Qwen-3. We also observe worse walltimes for the same block efficiencies. For example, Qwen-3 on ToolBench sees essentially the same walltimes for $K = 1$ and $K = 2$ as Llama-3 on GSM8K, even though Llama-3 has significantly higher block efficiencies than Qwen-3 (around $1.9$ and $1.7$ higher tokens/$M_p$-call for $K = 1$ and $K = 2$). Differences in hardware setups may additionally explain and influence these relative target and draft costs.

## J    TEMPERATURE ABLATIONS

To measure the impact of temperature on walltime speedups in GBV, we evaluate GBV on Qwen-3 32B/0.6B for GSM8K (500 prompts) with temperatures 0.2, 0.4, 0.6, 0.8, also including the temperature 1.0 run from Table 3. Our results are shown in Table 5.

Importantly, we find that GBV performs better at low temperature settings, which directly opposes block verification trends. Whereas standard block verification (the $K = 1$ rows) walltimes increase from under 80 ms/token to over 90 ms/token from temperatures 0.2 to 1.0, GBV with $K = 3$ sees its walltimes decrease from over 100 ms/token at temperature 1.0 to under 86 ms/token for lower temperatures (0.2, 0.4, 0.6), with its highest throughput at temperature 0.6. These results also show that temperature 1.0 sampling achieves over 10% lower throughput than all other GBV $K = 3$ settings, so our results in Section 6 actually reflect GBV in its worst-performing setting.

Even as we noted in Appendix I and Table 3 that standard BV may outperform GBV for Qwen-3 32B/0.6B, this finding no longer holds at low temperatures. For example, GBV with $K = 3$ achieves a throughput of around 5.6% higher tokens/s than BV when the target temperature is 0.2. Even at slightly higher temperatures of 0.4 and 0.6, GBV achieves a higher block efficiency than BV, and achieves similar throughput. Thus, even for model pairs where GBV with $K > 1$ does not outperform BV at temperature 1.0 sampling, it may still be preferable over BV at low temperatures.

| $K$ | $M_p$ Temp. | Block Efficiency (tokens/$M_p$-call) | Throughput (tokens/s) | Walltime (ms/token) |
|---|---|---|---|---|
| 1 | 0.2 | 3.867 | 11.034 | 90.626 |
| 1 | 0.4 | 4.313 | 12.407 | 80.598 |
| 1 | 0.6 | 4.308 | 12.368 | 80.851 |
| 1 | 0.8 | **4.430** | **12.761** | **78.362** |
| 1 | 1.0 | 4.322 | 12.557 | 79.638 |
| 3 | 0.2 | 4.219 | 11.661 | 85.759 |
| 3 | 0.4 | 4.333 | 11.911 | 83.954 |
| 3 | 0.6 | **4.492** | **12.149** | **82.311** |
| 3 | 0.8 | 4.161 | 11.341 | 88.173 |
| 3 | 1.0 | 3.571 | 9.946 | 100.547 |

Table 5: We evaluate how temperature impacts the block efficiency, throughput, and walltime for GBV with $K = 1, 3$ when run on Qwen-3 32B/0.6B with GSM8K (500 prompts) and $L = 8$. We use temperature 0.2, 0.4, 0.6, 0.8, and 1.0 sampling from the target model. While BV ($K = 1$) performs best at high temperatures (0.8, 1.0), GBV ($K = 3$) works better at low temperatures (0.2, 0.4, 0.6).

## K   DRAFT LENGTH ABLATIONS

Here, we provide ablation experiments on $L$, the draft block length. We extend our $L = 8$ results for Qwen-3 32B/0.6B on GSM8K with temperature 1.0 target sampling to larger draft lengths: $L = 16$ and $L = 24$. Our results are shown in Table 6.

As $L$ increases, block efficiency improves only modestly, while throughput degrades rapidly. For example, at $K = 1$, the block efficiency increases from 4.32 for $L = 8$ to 5.67 and 5.79 for $L = 16$ and $L = 24$, but the walltime grows from roughly 80 ms/token to 103 ms/token and 140 ms/token. The effect is worse for $K \geq 2$, where moving from $L = 8$ to $L = 24$ more than doubles the walltime.

The draft and target costs also shift substantially: the draft time, as a percentage of the end-to-end walltime, grows from about 71% at $L = 8$ to 80% and 86% at $L = 16, 24$, while the target share drops to as low as 10%. Coupled with our analysis of the draft bottleneck in Appendix I, this indicates longer blocks increase aggregate cost by pushing more work into the draft bottleneck.

| $L$ | $K$ | **Block Efficiency** (tokens/$M_p$-call) | **Walltime** (ms/token) | **Draft Cost** (%walltime) | **Target Cost** (%walltime) |
|---|---|---|---|---|---|
| 8 | 1 | **4.322** | **79.638** | **71.5** | **21.3** |
| | 2 | 3.910 | 92.823 | 71.3 | 21.0 |
| | 3 | 3.571 | 100.547 | 71.1 | 21.1 |
| | 4 | 3.429 | 106.123 | 70.8 | 21.3 |
| 16 | 1 | **5.667** | **102.970** | **81.7** | **13.2** |
| | 2 | 4.016 | 149.775 | 81.0 | 13.3 |
| | 3 | 3.915 | 155.406 | 80.5 | 13.6 |
| | 4 | 3.742 | 163.200 | 80.1 | 13.9 |
| 24 | 1 | **5.794** | **140.411** | **86.0** | **9.9** |
| | 2 | 4.275 | 196.937 | 85.0 | 10.2 |
| | 3 | 3.945 | 218.631 | 84.3 | 10.8 |
| | 4 | 3.628 | 240.142 | 83.4 | 11.5 |

Table 6: We evaluate GBV with $1 \leq K \leq 4$ for Qwen-3 32B/0.6B temperature 1.0 sampling on GSM8K (500 prompts), over varying draft block lengths $L = 8, 16, 24$. We report block efficiency, walltime, and draft and target forward pass runtimes as percentages of end-to-end walltime. We find that around $L = 8$ is an ideal length for avoiding more pronounced draft bottlenecks.

## L   MULTI-PATH COMPARISON

To empirically compare GBV to other multi-path methods, we implement six other speculative decoding algorithms in our code: standard speculative decoding (Leviathan et al., 2023; Chen et al., 2023), block verification (Sun et al., 2024b), NSS (Miao et al., 2024), SpecInfer (Miao et al., 2024), SpecTr (Sun et al., 2023), and traversal verification (Weng et al., 2025). SpecInfer provably improves NSS and block verification provably improves standard speculative decoding, so we only benchmark GBV against SpecInfer, SpecTr, and block verification.

We run comparison experiments for Llama-3 70B/8B on GSM8K (the same 500 prompts as previous experiments) for $L = 8$. We test $K = 2, 3$ as well as temperature 0.2, 1.0 sampling from the target model. Our block efficiency, throughput, and walltime numbers are shown in Table 7. Due to poor preliminary decoding rates and Llama-3 70B memory footprint, we do not run SpecTr and SpecInfer, the worst-performing temperature 1.0 algorithms, at temperature 0.2.

For temperature 1.0 sampling, GBV and traversal verification perform significantly better than SpecTr and SpecInfer. For both $K = 2, 3$, GBV achieves over 1.1 token/s higher throughput than SpecTr and over 1.6 token/s higher than SpecInfer. While traversal verification does outperform GBV in these settings, coming out on top, the incremental gains are less significant: GBV achieves around 0.6 fewer token/s for $K = 2$ and 0.9 fewer token/s for $K = 3$.

| $M_p$ Temp. | $K$ | Method | Block Efficiency (tokens/$M_p$-call) | Throughput (tokens/s) | Walltime (ms/token) |
|---|---|---|---|---|---|
| 1.0 | 2 | GBV | 4.430 | 7.508 | 133.188 |
| | | SpecTr | 3.716 | 6.355 | 157.365 |
| | | SpecInfer | 3.259 | 5.603 | 178.480 |
| | | **Traversal** | **4.760** | **8.114** | **123.245** |
| 1.0 | 3 | GBV | 4.271 | 7.338 | 136.272 |
| | | SpecTr | 3.540 | 6.211 | 161.002 |
| | | SpecInfer | 3.239 | 5.686 | 175.879 |
| | | **Traversal** | **4.779** | **8.260** | **121.065** |
| 0.2 | 2 | **GBV** | **6.730** | **11.415** | **87.602** |
| | | Traversal | 5.650 | 9.773 | 102.319 |
| 0.2 | 3 | **GBV** | **5.663** | **9.775** | **102.299** |
| | | Traversal | 5.763 | 9.962 | 100.379 |

Table 7: We compare GBV, SpecTr, SpecInfer, and traversal verification for Llama-3 70B/8B on GSM8K (500 prompts). We report block efficiency and throughput (higher is better), and walltime (lower is better), for $K = 2, 3$ and temperature $0.2, 1.0$ sampling from the target model. While traversal verification outperforms other methods for temperature 1.0, GBV achieves over 1.5 tokens/s higher throughput at temperature 0.2.

On the other hand, at lower temperatures like 0.2, the trend flips: GBV now performs better than traversal verification. While throughput and block efficiency gains are negligible for $K = 3$ (only around 0.1 more tokens/$M_p$-call and 0.2 higher token/s), the benefits for $K = 2$ are significant: GBV achieves with over 1.6 higher token/s than traversal verification, and over 1.1 higher tokens/$M_p$-call. In fact, GBV at temperature 0.2 with $K = 2$ achieves the highest throughput by far out of all tested settings: no other method for Llama-3 70B/8B achieves over 10 tokens/s throughput. This suggests that GBV outperforms traversal verification for low temperatures and sharper $p$ distributions.

