# OpenReview forum: "Greedy Multi-Path Block Verification for Faster Decoding in Speculative Sampling"
_ICLR.cc/2026/Conference — Submitted to ICLR 2026_

### Official Review · Reviewer_FQHP · 2025-10-20

**Soundness:** 2
**Presentation:** 2
**Contribution:** 2
**Rating:** 4
**Confidence:** 4

**Summary:**

The paper studies verification algorithms for L-step speculative decoding. It formulates an information-agnostic linear program (LP) and proves that Block Verification (BV) is optimal among all single-path verification schemes.  It then extends the LP to multi-path drafting. Several experiments on the OPT series model demonstrate the effectiveness of the method.

**Strengths:**

- The information-agnostic LP cleanly characterizes feasible node budgets with a theoretical guarantee.
- The appendices provide solid derivations and decomposition lemmas.
- This paper is overall well-written and easy to follow.

**Weaknesses:**

-  Results only use OPT models and three academic datasets; evaluation on modern LLMs (e.g., Llama-2/3/4 and Qwen2.5/3 families) and diverse tasks (long-context, multilingual, tool-use) would strengthen external validity.
- K=4 improves block efficiency but hurts wall-time (batch overhead dominates), which needs a more in-depth analysis of K to provide a deeper underrstanding of the method.
- Reproducibility would benefit from open code and configurations to validate GBV’s efficiency in other hardware.
- The author may want to porvide more  comparisons with strong multi-path or tree-based methods (e.g., staged/speculative variants, multi-token heads) and sensitivity to L, temperature, and draft quality.

**Questions:**

- Could you provide a systems breakdown (per-token FLOPs, KV-cache traffic, batch effects) to explain the K=4 wall-time regression and to guide deployment choices?
- How sensitive is GBV to temperature?
- Can GBV be combined with hierarchical or retrieval-augmented drafting and with multi-token heads?

---

> ### Author Response · Authors · 2025-11-28
> **Rebuttal by Authors (1/2)**
>
> Thank you for your detailed review on our work! We are glad that you found our information-agnostic LP to be a clean framework for node budgets, and that you found our presentation clear. We address your points below:
>
> **W1**
>
> We agree with your points on expanding the diversity of LLMs and tasks evaluated.
>
> We have updated our manuscript to include results for Llama-3 70B/8B and Qwen-3 32B/0.6B. For Qwen-3, we have also run global block verification on a wider range of datasets, now including a multilingual dataset (MGSM) as well as a tool-use dataset (ToolBench) alongside our three evaluated OPT datasets (GSM8K, MATH500, HumanEval). See W1 under our response to Reviewer 5zQa for our updated results.
>
> **W2**
>
> Thank you for raising this point. We agree that discussing the batch overhead from $K$ better frames our method, and we have updated our manuscript to include this discussion below.
>
> In our Llama-3 70B/8B experiments, we profiled the speculative loop on GSM8K and HumanEval for $K=1,2,3,4$ with $L=8$. The average draft tree size grows roughly linearly with $K$, being around $9,16,24,29$ at $K=1,2,3,4$. Across this $K$ sweep, the fraction of walltime allocated to the draft and target passes remains nearly constant (around 50% draft and 40% target), and the peak KV-cache footprint of the target model stays in a narrow band (roughly 80 MB on GSM8K and 110 MB on HumanEval, with the draft cache around 40 MB). This shows that increasing $K$ mainly scales up FLOPs and KV traffic in proportion to the larger tree, rather than changing the bottleneck. Thus, although block efficiency is reasonably high, end-to-end walltime suffers. For this model pair, the overhead of batched forward passes dominates, so the optimal deployment choice is to keep $K=1$ (single-path block verification).
>
> For Qwen-3 32B/0.6B, we performed the same profiling on five datasets (GSM8K, HumanEval, MATH500, MGSM, ToolBench) for $K=1,2,3,4$ at $L=8$. Again, the average draft-tree size per target call grows almost linearly with $K$. The draft and target pass runtimes also remain nearly constant relative to the walltime. However, now, over 70% of time is spent in the draft stage and only around 20% in the target stage. The peak KV-cache footprint also stays in a narrow band as $K$ increases, with the target ranging from roughly 60 to 120 MB across datasets and the draft from around 30 to 50 MB, with only small variation across $K$. Again, increasing $K$ scales work in proportion to the tree size, but does not fundamentally shift where time is spent: the drafting dominates the cost for this model pair. Combined with the fact that block efficiency actually decreases as $K$ grows (on GSM8K and MGSM it drops from about 4.3 at $K=1$ to about 3.4 at $K=4$), this explains why GBV with $K>1$ is not particularly effective for temperature one sampling on Qwen-3 32B/0.6B.
>
> **W3**
>
> We agree with your point on reproducibility. We have open-sourced our anonymous code with system profiling at the following link: (https://anonymous.4open.science/r/Greedy_Block_Verification-B800). This includes not only our implementation of GBV and evaluation framework, but also the implementations of six other speculative decoding algorithms (standard, Block Verification, Naive Speculative Sampling, SpecInfer, SpecTr, Traversal Verification) which we compared GBV against. See the end of W4 for the comparison against the best-performing methods.
>
> **W4**
>
> We provide results for $L$ experiments below, extending the $L=8$ paper setting to $L=16,24$. These block efficiency and walltime numbers are for Qwen3-32B/Qwen3-0.6B on GSM8K.

---

> ### Author Response · Authors · 2025-11-28
> **Rebuttal by Authors (2/2)**
>
> **W4 (continued)**
>
> As $L$ increases, block efficiency improves only modestly, while walltime degrades rapidly. For example, at $K=1$ the block efficiency increases from $4.32$ for $L=8$ to $5.67$ and $5.79$ for $L=16$ and $L=24$, but the walltime grows from roughly 80 ms/token to 103 ms/token and 140 ms/token. The effect is more pronounced for $K \geq 2$: moving from $L=8$ to $L=24$ nearly doubles the walltime. The draft/target time breakdown also shifts: the draft time (relative to full walltime) grows from about 71% at $L=8$ to 80% and 86% at $L=16,24$, while the target share drops to as low as 10%, indicating that longer draft blocks increase cost by pushing more work into the draft bottleneck.
>
> | L  | K | Block eff (tokens / $M_p$-call) | Walltime (ms/token) | Draft time (%) | Target time (%) |
> | -- | - | ---------------------------- | ------------------- | -------------- | --------------- |
> | 8  | 1 | 4.322                        | 79.638              | 71.5           | 21.3            |
> | 8  | 2 | 3.910                        | 92.823              | 71.3           | 21.0            |
> | 8  | 3 | 3.571                        | 100.547             | 71.1           | 21.1            |
> | 8  | 4 | 3.429                        | 106.123             | 70.8           | 21.3            |
> | 16 | 1 | 5.667                        | 102.970             | 81.7           | 13.2            |
> | 16 | 2 | 4.016                        | 149.775             | 81.0           | 13.3            |
> | 16 | 3 | 3.915                        | 155.406             | 80.5           | 13.6            |
> | 16 | 4 | 3.742                        | 163.200             | 80.1           | 13.9            |
> | 24 | 1 | 5.794                        | 140.411             | 86.0           | 9.9             |
> | 24 | 2 | 4.275                        | 196.937             | 85.0           | 10.2            |
> | 24 | 3 | 3.945                        | 218.631             | 84.3           | 10.8            |
> | 24 | 4 | 3.628                        | 240.142             | 83.4           | 11.5            |
>
> For temperature experiments, see W3 in our response to Reviewer 5zQa.
>
> We note draft quality ablations are implicitly reflected in the model family and the target/draft size discrepancy. We feel that our new model experiments (see W1) adequately cover the former, and our original experiments with different OPT drafts address the latter.
>
> Under Q2 in our response to Reviewer i8ax, we have also added comprehensive coverage and comparison of our approach to the strongest multi-path methods. While you mentioned staged/speculative variants and multi-token heads as potential points of comparison, we believe that three other methods are the most reasonable points of comparison: Traversal Verification (https://arxiv.org/pdf/2505.12398, mentioned by Reviewer i8ax), SpecInfer (https://arxiv.org/abs/2305.09781), and SpecTr (https://arxiv.org/abs/2310.15141). Staged variants (e.g. Staged Speculative Decoding: https://arxiv.org/pdf/2308.04623) and multi-token heads (e.g. Medusa: https://arxiv.org/pdf/2401.10774) mainly alter the drafting phase rather than verification algorithm. On the other hand, Traversal Verification, SpecInfer, and SpecTr, like our GBV method, are entirely new verification algorithms which make no changes to or assumptions about drafting, and so they are the most adjacent to our work.
>
> **Q1**
>
> We have provided a systems breakdown on $K=4$ walltime regression under W2 above and the effects of $L$ (draft block length) in W4 above, to further explain how one would optimally choose $K$ and $L$ in real deployment settings.
>
> **Q2**
>
> GBV is highly sensitive to the target distribution temperature. In fact, while block verification tends to perform worse at lower temperatures, GBV actually performs significantly better. Thus, the temperature one results in our original manuscript frame GBV in its worst-performing setting. See W3 in our response to Reviewer 5zQa for more discussion on the impact of temperature.
>
> **Q3**
>
> Yes, GBV can be combined with any hierarchical or retrieval-augmented drafting method, as well as multi-token heads. These mechanisms only alter the underlying drafting phase, i.e. how the draft tree is formed. On the other hand, GBV aims to only improve the verification phase, i.e. how we select and throw out tokens from the draft tree. As long as GBV takes in the true probabilities used to sample the draft tree, it can be used with any drafting mechanism.
>
> We thank you again for your in-depth analysis of our work and for your strong feedback on experimental improvements, multi-path comparisons, and system profiling. We have made a significant effort to implement and incorporate all of your suggestions into our work, as well as open-source our main algorithm and all benchmarks. Please let us know if you have any lingering doubts about our results or responses, or further ideas on how to improve our work.

---

### Official Review · Reviewer_i8ax · 2025-11-01

**Soundness:** 3
**Presentation:** 2
**Contribution:** 2
**Rating:** 4
**Confidence:** 3

**Summary:**

The paper formalizes verification for speculative decoding via an information‑agnostic LP. It shows (i) in the single‑path case, block verification (BV) remains optimal even if the verifier is allowed full off‑path probabilities; and (ii) in the multi‑path case (K>1), off‑path information can help, leading to a decomposition into path selection plus single‑path BV. Building on this, the authors propose a practical, on‑path‑only greedy multi‑path block verification (GBV) and report sizable block‑efficiency gains with best wall‑clock improvements at K=3.

**Strengths:**

Clear formalization and optimality in single‑path. The LP view cleanly isolates prefix‑matching as the true bottleneck. Theorem 3.3 and Theorem 3.4 together pin down BV as optimal among all valid single‑path algorithms, not just on‑path‑restricted ones.
Multi‑path decomposition. The paper’s factorization, randomized path selection to induce a “skewed draft” followed by single‑path BV, is conceptually neat and technically useful for designing approximations.
Practical algorithm & evidence. GBV is implementable using only on‑path probabilities. Experiments show monotone block‑efficiency gains with K (diminishing returns) and best wall‑times at K=3, with K=4 hurting latency, is an honest and useful practical takeaway.

**Weaknesses:**

1. Related‑work coverage needs to be tightened and comparisons made precise.

The paper states that tree verification (Hu & Huang, 2024) improves over token‑wise verification but is provably worse than block verification (BV). Sun et al. (2024b) prove BV’s optimality among single‑path, on‑path verification algorithms, but that result does not by itself imply a strict separation from tree verification. If this is the intended positioning, please give a crisp statement of assumptions and regimes and, if possible, a short proof sketch or a pointer to a lemma that implies the strict separation (the current text only states it).

I could not find a citation or discussion of “Traversal Verification for Speculative Tree Decoding”, which appears closely related to verification on drafted trees and off‑path usage. Given your multi‑path/tree motivations, a comparison seems important, please add and position it.

2. Strength of the verification‑class constraints should be better motivated.

Definition 3.1 enforces single‑path prefix‑matching by construction, your own results show this is exactly why single‑path optimality caps out at BV even with full off‑path information. It would help to explicitly foreground this as a modeling choice (not just a technical convenience) and to discuss whether any practically relevant relaxations of prefix‑matching are possible or desirable.  For example, “Traversal Verification for Speculative Tree Decoding” takes a less restrictive choice.

3. typos:

Theorem 4.6, Equation (17): missing a summation on the LHS.
Equations (22), (23): the conditional distributions are omitted in the notation.

**Questions:**

Comparison to “Traversal Verification for Speculative Tree Decoding.” How does GBV (path selection + BV) compare, conceptually and empirically, to traversal‑based verification over a draft tree? At $L=1$ special case, does “Traversal Verification for Speculative Tree Decoding” give better acceptance rate than GBV?

---

> ### Author Response · Authors · 2025-11-28
> **Rebuttal by Authors (1/2)**
>
> Thank you for your comprehensive review! We appreciate that you found our single-path LP formulation and factorization into path selection to be comprehensive and clear. We address your points below:
>
> **W1**
>
> We agree with your points on tightening related-work coverage. We discuss your points on Hu and Huang below, and we have updated our manuscript to include this discussion.
>
> While tree verification from Hu and Huang (https://openreview.net/pdf?id=stMhi1Sn2G) may appear to be separate from the single-path, on-path regime of block verification (BV), it actually falls under the same category: it is not truly a multi-path method. To see this, we refer to the end of Section 5 in Hu and Huang, where it is explicitly stated that the draft model sequentially samples $d$ reference tokens from the draft model, like in block verification, forming a draft chain rather than a draft tree.
>
> Our claim that tree verification falls under the BV regime, and is thus suboptimal compared to BV, is supported by the BV paper (https://arxiv.org/pdf/2403.10444): “The only other work that we are aware of that improves the verification step in speculative decoding is the independent work of Hu and Huang (2024), which uses tree Monte Carlo to improve speculative decoding in the single draft case, and have proved that their algorithm improves over token verification. On the contrary, we prove that our algorithm achieves the optimal speedup among all valid verification algorithms, including theirs.”
>
> Also, thank you for bringing the Traversal Verification work (https://arxiv.org/pdf/2505.12398) to our attention! We were not aware of this work at submission time, but have since updated our manuscript to include relevant discussion. Traversal Verification is motivated by turning the typical top-down approach to verification into a bottom-up approach. Nevertheless, it is still a valid information-restricted multi-path verification algorithm under our framework. In fact, just like our greedy block verification approach, the authors mention Traversal Verification reduces to BV (the optimal single-path algorithm) for $K=1$ drafted paths: “Since Block Verification is an optimal valid chain verification algorithm [27, Theorem 2], Traversal Verification inherits this optimality in the single-chain case (see Theorem 3.4).” For $K>1$ paths, Traversal Verification and greedy block verification may be better suited for different target/draft distribution regimes.
>
> **W2**
>
> Thank you for raising this point. We agree that it is important to motivate the prefix-matching as a modeling choice, and not just a technical convenience. We do this below, and have updated our manuscript to include this discussion.
>
> Prefix-matching is necessary due to the nature of the autoregressive token generation in LLMs. One cannot access the next-token target model distribution $p(\cdot | c)$ for nodes (contexts) $c$ that lie outside the draft tree (which are necessary to sample tokens that respect the target distribution) without performing another target model forward pass. This extra pass would defeat the goal of speculative decoding: decoding multiple tokens with a single target model call. Hu and Huang (https://openreview.net/pdf?id=stMhi1Sn2G) explain this in further detail at the beginning of their Section 5 following: “An additional requirement for speculative sampling is that the sampling process must respect the inherent tree structure.”
>
> We do believe that there are potential avenues for future work which relax prefix-matching, say by removing the requirement that the output must strictly match the target distribution. But these are not true speculative decoding algorithms, and thus lie outside our scope.
>
> On your point regarding Traversal Verification (https://arxiv.org/pdf/2505.12398), we believe that it actually does not take a less restrictive choice, as it still satisfies target-matching and prefix-matching under our framework. See Algorithm 3 of their paper: the output $X^{\tau},Y$ (line 17) must satisfy that $X^{\tau}$ (line 7) is a chain $\alpha$ in the draft tree $T$ (line 4), meaning that the output with its last token removed is a node on the draft tree. This is exactly the prefix-matching requirement. Target-matching is a necessary requirement for a valid speculative decoding algorithm, and they prove this property in Theorem 3.3.
>
> **W3**
>
> Thank you for catching the typo in Theorem 4.6! We have updated the left hand side to include the summation.
>
> For Equations (22) and (23), we have updated the manuscript to clarify that the conditional distribution over the current context $c$ is implicit in all $q$ notation.

---

> ### Author Response · Authors · 2025-11-28
> **Rebuttal by Authors (2/2)**
>
> **Q1**
>
> We address the theoretical comparison to Traversal Verification in W1 and W2 above.
>
> **Q2**
>
> In the $L=1$ setting, similar to our point at the end of W1, we expect Traversal Verification would again be suited for different target/draft regimes than our approach: there is no clear theoretical indication as to which method performs better. Accordingly, we have implemented Traversal Verification alongside two multi-path methods, SpecInfer (https://arxiv.org/abs/2305.09781) and SpecTr (https://arxiv.org/abs/2310.15141), and run comparison experiments against GBV. Our results for Llama-3 70B/8B on GSM8K with $L=8$ are shown below, with temperature 1.0 sampling from the target model.
>
> We see that GBV and Traversal Verification both perform significantly better than SpecTr and SpecInfer, with each achieving over one token/s higher throughput than both. Furthermore, Traversal Verification performs better than GBV, although the throughput gains are smaller for $K=2$ (around 0.6 token/s higher).
>
>
> | Method    | K | Block eff. (tokens / $M_p$-call) | Throughput (tokens/s) | Walltime (ms/token) |
> | --------- | - | ----------------------------: | --------------------: | ------------------: |
> | GBV       | 2 |                         4.430 |                 7.508 |             133.188 |
> | SpecTr    | 2 |                         3.716 |                 6.355 |             157.365 |
> | SpecInfer | 2 |                         3.259 |                 5.603 |             178.480 |
> | Traversal | 2 |                         4.760 |                 8.114 |             123.245 |
> | GBV       | 3 |                         4.271 |                 7.338 |             136.272 |
> | SpecTr    | 3 |                         3.540 |                 6.211 |             161.002 |
> | SpecInfer | 3 |                         3.239 |                 5.686 |             175.879 |
> | Traversal | 3 |                         4.779 |                 8.260 |             121.065 |
>
>
> However, it is not true that Traversal Verification performs better than GBV in all settings. As mentioned in W3 of our response to Reviewer 5zQa, GBV performs better when the target model temperature is low. Below, we set the temperature to 0.2 (all other experimental settings held the same, e.g. run Llama-3 70B/8B on GSM8K), and we see that GBV outperforms Traversal Verification by over 1.5 higher token/s of throughput when $K=2$, and only 0.2 lower token/s when $K=3$. Since GBV at $K=2$ achieves the highest block efficiency and throughput and lowest walltime by far in this setting, this suggests that GBV outperforms Traversal Verification for lower temperatures and sharper $p$ distributions.
>
> | Method    | K | Block eff. (tokens / $M_p$-call) | Throughput (tokens/s) | Walltime (ms/token) |
> |-----------|---|-------------------------------:|----------------------:|--------------------:|
> | GBV       | 2 | 6.730                         | 11.415                | 87.602             |
> | Traversal | 2 | 5.650                         | 9.773                 | 102.319            |
> | GBV       | 3 | 5.663                         | 9.775                 | 102.299            |
> | Traversal | 3 | 5.763                         | 9.962                 | 100.379            |
>
>
> Thank you again for bringing the Traversal Verification work to our attention, and for your thoughtful feedback on the presentation and motivation behind our definition of valid multi-path verification algorithms. By incorporating your suggestions, we have significantly improved the contextualization of our work in broader speculative decoding literature, and demonstrated the benefits of our method over strong existing multi-path works. If you have any questions on any of our responses, or further suggestions on how to improve our work, please let us know.

---

### Official Review · Reviewer_5zQa · 2025-11-04

**Soundness:** 3
**Presentation:** 3
**Contribution:** 3
**Rating:** 4
**Confidence:** 3

**Summary:**

This paper proposes Greedy Multi-Path Block Verification (GBV), a practical extension of block verification for speculative decoding.
The authors formulate an information-agnostic linear program that characterizes all valid verification algorithms under prefix- and target-matching constraints, showing that standard block verification remains optimal in the single-path case. They then extend this formulation to multiple draft paths (K > 1), derive submodular constraints, and develop a tractable greedy approximation using tree-based path ranking with only on-path probabilities. Empirically, GBV improves block efficiency by 30% and reduces decoding walltime by around 15% on OPT model pairs across GSM8K, HumanEval, and MATH500.

**Strengths:**

The paper writing is clear. The paper provides detailed theoretical analysis.  The paper also presents an efficient method called
greedy multi-path block verification.

**Weaknesses:**

The evaluation is limited. First, this paper only select block efficiency and wall time as metrics. For other metrics, such as acceptance rates should also be incorporated. Moreover, this paper only select one model family, such as OPT and only consider relatively small model such as 6.7B. It is necessary to investigate the effectiveness of proposed approach on larger models. Furthermore, this paper does not consider some important factors, e.g., temperature. Different temperature may lead to different distribution and it is not clear whether the proposed approach can be robust across all temperature settings.

**Questions:**

See weaknesses.

---

> ### Author Response · Authors · 2025-11-28
> **Rebuttal by Authors (1/2)**
>
> Thank you for your detailed review of our work! We are glad that you found our theoretical analysis of GBV to be comprehensive and clear. We address your points below:
>
> **W1**
>
> Thank you for raising the important point that in addition to our main evaluation metrics of block efficiency and walltime speedup, criteria like acceptance rates should be incorporated.
>
> We would like to clarify that in previous works which introduce novel multi-path speculative decoding methods, including SpecTr (https://arxiv.org/pdf/2310.15141), SpecInfer (https://arxiv.org/pdf/2305.09781), and Traversal Verification (https://arxiv.org/pdf/2505.12398), block efficiency and walltime speedup are the primary evaluation criteria.
>
> Block efficiency is the average number of accepted tokens per call to the target model, so it is essentially an acceptance rate in the multi-step setting (and in fact some works refer to it as acceptance, not block efficiency). This is roughly the speedup of speculative decoding over vanilla decoding. Walltime numbers give the true speedup, which deviates from the block efficiency due to factors like the generation cost of a multi-path draft tree, and the increased cost of the target model forward pass over the tree, which also depend on the hardware setup.
>
> We would like to emphasize that there is no natural notion of single-token acceptance rates in greedy block verification. In other works like SpecTr, the proposed multi-path verification algorithm builds upon a single-step multi-draft algorithm (see line 1 of Algorithm 3 in SpecTr) called an OTLP (optimal transport linear program) solver, which takes in multiple tokens and probabilistically returns a single token. There is a notion of single-token acceptance rates in OTLP solvers: how often the output token lies among the input tokens. Works which improve SpecTr, like “Canonical Decomposition” (https://arxiv.org/pdf/2410.18234) and “Towards Optimal Multi-Draft Speculative Decoding” (https://arxiv.org/pdf/2502.18779), again deal with improving the OTLP solver, and therefore also include token-by-token acceptance rates as a metric.
>
> However, our work makes no use of an OTLP solver, so single-token acceptance cannot be used as a metric. This is similar to block verification (https://arxiv.org/pdf/2403.10444) and traversal verification (https://arxiv.org/pdf/2505.12398), who only report block efficiency and walltime speedup numbers. Because of this metric limitation, we have instead opted to include system-side metrics, such as KV cache traffic and a breakdown of the draft and target model forward pass costs, to explain where discrepancies between walltime numbers and block efficiencies might arise. For more details, see our response to reviewer FQHP under W2/W4.
>
> **W2**
>
> We agree that our evaluation should cover more models than just the OPT family, and incorporate larger model sizes. To incorporate your suggestions, we implement GBV for Qwen-3 32B/0.6B on five datasets (GSM8K, HumanEval, MATH500, MGSM, ToolBench) with $L=8$ and temperature one sampling. Our results are shown below.
>
> Interestingly, for this model pair, the best setting is the single-path block verification baseline ($K=1$): on GSM8K, block efficiency is $4.32$ tokens per $M_p$-call at $K=1$, but decreases to $3.91$, $3.57$, and $3.43$ for $K=2,3,4$ respectively, while walltime increases from $79.6$ to $92.8$, $100.5$, and $106.1$ ms/token. Similar trends hold across HumanEval, MATH500, MGSM, and ToolBench.
>
> | Model pair      | Dataset   | K=1 (tokens / $M_p$-call) | K=2   | K=3   | K=4   |
> | --------------- | --------- | ---------------------- | ----- | ----- | ----- |
> | Qwen-3 32B/0.6B | GSM8K     | **4.322**              | 3.910 | 3.571 | 3.429 |
> | Qwen-3 32B/0.6B | HumanEval | **4.204**              | 3.423 | 3.331 | 2.960 |
> | Qwen-3 32B/0.6B | MATH500   | **4.342**              | 3.917 | 3.693 | 3.411 |
> | Qwen-3 32B/0.6B | MGSM      | **4.329**              | 3.973 | 3.468 | 3.445 |
> | Qwen-3 32B/0.6B | ToolBench | **3.090**              | 2.706 | 2.791 | 2.681 |
>
> | Model pair      | Dataset   | K=1 (ms/token) | K=2     | K=3     | K=4     |
> | --------------- | --------- | -------------- | ------- | ------- | ------- |
> | Qwen-3 32B/0.6B | GSM8K     | **79.638**     | 92.823  | 100.547 | 106.123 |
> | Qwen-3 32B/0.6B | HumanEval | **84.166**     | 106.051 | 109.215 | 124.554 |
> | Qwen-3 32B/0.6B | MATH500   | **81.517**     | 91.133  | 97.880  | 106.575 |
> | Qwen-3 32B/0.6B | MGSM      | **80.992**     | 90.312  | 102.296 | 105.387 |
> | Qwen-3 32B/0.6B | ToolBench | **113.563**    | 132.477 | 128.405 | 133.973 |

---

> ### Author Response · Authors · 2025-11-28
> **Rebuttal by Authors (2/2)**
>
> **W2 (continued)**
>
> To test even larger target and draft model sizes, we implement GBV for Llama-3 70B/8B on two datasets (GSM8K, HumanEval). Again, $K=1$ appears to be the optimal setting. Nevertheless, unlike the Qwen results, there can be meaningful gains at higher $K$: on GSM8K, the walltime decreases by nearly 10% from $K=3$ to $K=4$.
>
> | Model pair     | Dataset   | K=1 (tokens / $M_p$-call) | K=2  | K=3   | K=4   |
> | -------------- | --------- | ---------------------- | ----- | ----- | ----- |
> | Llama-3 70B/8B | GSM8K     | **4.943**              | 4.430 | 4.271 | 4.625 |
> | Llama-3 70B/8B | HumanEval | **5.993**              | 4.806 | 4.473 | 4.370 |
>
> | Model pair     | Dataset   | K=1 (ms/token) | K=2     | K=3     | K=4     |
> | -------------- | --------- | -------------- | ------- | ------- | ------- |
> | Llama-3 70B/8B | GSM8K     | **114.123**    | 133.188 | 136.272 | 123.545 |
> | Llama-3 70B/8B | HumanEval | **96.063**     | 124.540 | 131.889 | 135.186 |
>
> We view these results as an example of our framework guiding deployment choices. GBV shows that for some model pairs (our two original OPT pairs) $K=2,3$ are optimal, whereas for others (Qwen-3 32B/0.6B and Llama-3 70B/8B) the best choice is to revert to $K=1$ (standard BV).
>
>
>
>
> **W3**
>
> Thank you for emphasizing the importance of temperature variation. We have now run GBV with Qwen-3 32B/0.6B on the GSM8K dataset with temperatures $p_{temp}=0.2, 0.4, 0.6, 0.8,1.0$, for target model sampling (draft temperature is kept at 1.0). Our results are shown below.
>
> | K | $p_{temp}$ | Block eff (tokens / $M_p$-call) | Throughput (tokens/s) | Walltime (ms/token) |
> | - | ------ | ---------------------------- | --------------------- | ------------------- |
> | 1 | 0.2    | 3.867                        | 11.034                | 90.626              |
> | 1 | 0.4    | 4.313                        | 12.407                | 80.598              |
> | 1 | 0.6    | 4.308                        | 12.368                | 80.851              |
> | 1 | 0.8    | 4.430                        | 12.761                | 78.362              |
> | 1 | 1.0    | 4.322                        | 12.557                | 79.638              |
> | 3 | 0.2    | 4.219                        | 11.661                | 85.759              |
> | 3 | 0.4    | 4.333                        | 11.911                | 83.954              |
> | 3 | 0.6    | 4.492                        | 12.149                | 82.311              |
> | 3 | 0.8    | 4.161                        | 11.341                | 88.173              |
> | 3 | 1.0    | 3.571                        | 9.946                 | 100.547             |
>
> **Importantly, we find that greedy block verification performs better at low temperature settings, which directly opposes block verification trends.** Whereas standard BV ($K=1$ rows above) has decreasing throughput and increased walltimes as temperature decreases, GBV ($K=3$ rows above) actually sees its highest throughputs at lower temperatures ($0.2,0.4,0.6$). These results also show that temperature one sampling achieves over 10% lower throughput than all other GBV settings, so our numbers from W2 actually reflect GBV in its worst-performing setting.
>
> Also, even as we mentioned in W2 that Qwen-3 32B/0.6B is a model pair where $K=1$ GBV (standard BV) may outperform $K>1$ GBV, this no longer holds at low temperatures. For example, GBV with $K=3$ achieves a throughput of around 5.6% higher tokens/s than BV when the target temperature is 0.2. Even at slightly higher temperatures of 0.4 and 0.6, GBV achieves a higher block efficiency than BV, and similar throughput.
>
> **Thus, even for model pairs where GBV with $K>1$ does not outperform BV at temperature one target sampling, GBV can still be far more effective for low temperatures.**
>
> Again, thank you for your thorough feedback and insightful suggestions on evaluation criteria, which have allowed us to significantly improve upon the empirical soundness and efficacy of our multi-path greedy block verification method. If you have any questions on any of our responses, or further suggestions on how to improve our work, please let us know.

---

### Author Response · Authors · 2025-12-02
**Meta-Rebuttal by Authors**

We thank the reviewers for their detailed suggestions on how to improve our work. We note that all reviewers highlighted the efficacy of our theoretical LP framework and found our greedy block verification (GBV) algorithm to be effective. We have directly addressed all feedback from reviewers, including additional experiments and ablations, which we summarize below.

In response to reviewers **5zQa** and **FQHP**, who raised concerns about the the coverage of our GBV evaluations, we ran additional experiments for **new model families** (Qwen-3 and Llama-3) , **multilingual and tool-use datasets** (MGSM and ToolBench for Qwen-3), and **much larger target models** (Qwen-3 32B and Llama-3 70B). We found that while GBV with $K>1$ is highly effective for OPT models, for temperature 1.0 sampling in Qwen-3 and Llama-3, the best option is to revert to GBV with $K=1$ (standard block verification).

Reviewers **5zQa** and **FQHP** asked about the effectiveness of GBV under **different temperatures**, as all our original experiments were for temperature 1.0 sampling from the target model. Accordingly, we ran ablations for Qwen-3 32B/0.6B on GSM8K with temperatures 0.2, 0.4, 0.6, and 0.8. We found that GBV with $K>1$ performs better than GBV with $K=1$ (standard block verification) at low temperatures, the opposite trend to our Qwen-3 results at temperature 1.0. Thus, our approach is most effective at low temperatures when the target distribution is sharp, and our original results actually frame GBV in its worst setting.

Reviewers **5zQa** and **FQHP** also wanted **metrics other than block efficiency and wall time**. In our response to reviewer **5zQa**, we explained why acceptance rates, which are reported in OT-based verification works like SpecTr and SpecInfer, do not apply to our GBV setting (and are also not reported in non-OT works like block verification). Then, in our response to reviewer **FQHP**, we profiled more suitable systems metrics for Llama-3 and Qwen-3. We discussed the cost of draft tree construction and the batched target forward pass relative to end-to-end wall time, the peak KV-cache footprint for target and draft models, and how draft tree size scales with $K$. These results reveal that increasing $K$ scales target and draft costs in proportion to the tree size, but does not fundamentally change the drafting bottleneck.

Reviewer **FQHP** also questioned **the impact of $L$**, the draft block length, as our original experiments were for $L=8$. We included additional ablations for Qwen-3 32B/0.6B on GSM8K on $L=16$ and $L=24$. We found that for higher $L$, the drafting stage occupies a larger portion of the end-to-end wall time (over 10% more), which leads to significant degradation in wall time even when block efficiency increases for higher $L$. Thus, GBV and other multi-path methods are likely most effective around the $L=8$ range.

Our $K,L$-profiling also addresses the request for a **systems breakdown** by reviewer **FQHP**, to explain $K=4$ wall time regression and guide $K,L$-selection during deployment.

Reviewer **i8ax** mainly asked for a **comparison to the “Traversal Verification” work** and **motivation for our verification constraints**. In our response, we clarified how other algorithms like Hu and Huang’s tree verification sit in our framework, why prefix-matching is necessary due to the nature of autoregressive LLM generation and caching, and why Traversal Verification falls under our framework. We also theoretically compared GBV to Traversal Verification, noting that both reduce to standard block verification for $K=1$, but are more effective in different target/draft settings. Even in the $L=1$ special case, there is no universal guarantee on which algorithm performs better.

Since reviewer **i8ax** also wanted an **empirical comparison between GBV and Traversal Verification**, and reviewer **FQHP** asked for a **comparison to other strong multi-path methods**, we addressed both of their points by implementing the six most relevant verification algorithms (naive, block verification, Naive Speculative Sampling, SpecInfer, SpecTr, Traversal Verification) to our GBV work, and we open-sourced our anonymized code (https://anonymous.4open.science/r/Greedy_Block_Verification-B800). Our results showed that for temperature 1.0 sampling from Llama-3 on GSM8K for $K=2,3$, the top method is Traversal Verification, followed closely by GBV, with SpecTr and SpecInfer lagging further behind. On the other hand, at lower temperatures like 0.2, GBV performs significantly better than Traversal Verification (over 1.5 higher tokens per second).

We thank the reviewers again for their detailed comments on evaluations, comparisons, and related works in the broader speculative decoding literature. These have led to significant improvements in our experimental coverage, stronger guidance on how to select system parameters, and key insights on when GBV outperforms state-of-the-art multi-path methods.

---

### Meta-Review · Area_Chair_z7tu · 2025-12-20

**Summary:**

**Paper summary.** This paper studies verification in speculative decoding and proposes an information-agnostic LP framework to characterize valid verification algorithms under prefix/target matching constraints. It proves optimality of block verification (BV) among single-path schemes and extends the view to a multi-path setting, leading to a greedy multi-path block verification (GBV) approach and a factorization into path selection + BV. The paper mixes theory (LP formulation and optimality statements) with system metrics (block efficiency and wall-time).

**What happened in the discussion.** All reviewers scored this as borderline (three 4’s) and the discussion focused on: (1) limited evaluation scope (initially mostly one model family and small models), (2) missing systems-level explanation for why larger K can hurt wall-time, and (3) related-work positioning (tree/traversal verification comparisons). In rebuttal, the authors added substantial additional evidence: experiments on larger and more modern model families (including Llama-3 70B/8B, Qwen-3), added datasets beyond the original set (including multilingual and tool-use datasets), provided detailed draft/target time breakdown tables, and implemented additional comparisons (including traversal verification, SpecInfer, SpecTr) with throughput results. They also clarified why block efficiency is used as the primary metric and how it relates to acceptance rate.

**My assessment as AC.** The LP formalization is clean and the rebuttal significantly strengthened the empirical side. Even so, the overall contribution is incremental relative to recent work in this area (the paper extends and sharpens known results more than it introduces a new algorithmic paradigm). Given this batch’s limited acceptance budget, I am leaning to reject, but I want to be clear that the paper is close: with a sharper positioning and a stronger “why this matters beyond this proof” story, it could be competitive.

**Decision.** Reject (borderline). This is mainly due to competitiveness and taste, not because the work is flawed. I encourage the authors to keep the broader evaluation and the wall-time breakdown, and to clearly state what is new beyond prior BV optimality results and how the multi-path factorization translates into practical guidance for real deployments.

**Reviewer Concerns:**

- **Reviewer FQHP (rating 4, confidence 4)**: Asked for broader model families/tasks and a clearer systems breakdown of when K hurts wall-time. Authors added experiments on additional model families and provided draft/target time breakdown and pipeline/batch overhead discussion. **Status:** mostly resolved.
- **Reviewer i8ax (rating 4, confidence 3)**: Requested tighter related-work positioning, especially vs. traversal/tree verification, and clearer conceptual comparison in L=1 special cases. Authors added discussion and implemented traversal verification comparisons. **Status:** partially resolved (positioning is improved but still dense).
- **Reviewer 5zQa (rating 4, confidence 3)**: Asked for broader evaluation and additional metrics; authors added broader evaluation and clarified metrics and interpretation. **Status:** mostly resolved.

**Reviewer Scores:**

- **FQHP (4,4)**: Could move up after the added experiments and breakdown, but no explicit score change is recorded.
- **i8ax (4,3)**: Likely unchanged.
- **5zQa (4,3)**: Likely unchanged.

---

### Decision · Program_Chairs · 2026-01-26

Reject